Semantic approaches for query expansion: taxonomy, challenges, and future research directions

Allahim Azzah 1 2 azzah.allahim@gmail.com
Cherif Asma 1 3
Imine Abdessamad 4
1 IT Department, King Abdulaziz University, Faculty of Computing and Information Technology , Jeddah , Saudi Arabia
2 College of Computer and Information Sciences, Jouf University , Sakaka , Saudi Arabia
3 The Center of Excellence in Smart Environment Research, King Abdulaziz University , Jeddah , Saudi Arabia
4 Université de Lorraine, CNRS, Inria, LORIA , Nancy , France
Alatas Bilal
Electronic publication date: 2025 Mar 5
Publication date: 2025
Volume: 11
Electronic Location ID: e2664
Received 2024 May 28; Accepted 2024 Dec 30
Copyright: © 2025 Allahim et al.
Copyright year: 2025
Copyright holder: Allahim et al.
License: This is an open access article distributed under the terms of the Creative Commons Attribution License, which permits unrestricted use, distribution, reproduction and adaptation in any medium and for any purpose provided that it is properly attributed. For attribution, the original author(s), title, publication source (PeerJ Computer Science) and either DOI or URL of the article must be cited.
License URL: https://creativecommons.org/licenses/by/4.0/

Keywords: Query expansion, Semantic query, Semantic web, Ontology

Funding: The authors received no funding for this work.

==============================
The internet has been inundated with an ocean of information, and hence, information retrieval systems are failing to provide optimal results to the user. In order to meet the challenge, query expansion techniques have emerged as a game-changer and are improving the results of information retrieval significantly. Of late, semantic query expansion techniques have attracted increased interest among researchers since these techniques offer more pertinent and practical results to the users. These allow the user to retrieve more meaningful and useful information from the web. Currently, few research works provide a comprehensive review on semantic query expansion; usually, they cannot provide a full view on recent advances, diversified data application, and practical challenges. Therefore, it is imperative to go deep in review in order to explain these advances and assist researchers with concrete insights for future development. This article represents the comprehensive review of the query expansion methods, with a particular emphasis on semantic approaches. It overviews the recent frameworks that have been developed within a period of 2015–2024 and reviews the limitations of each approach. Further, it discusses challenges that are inherent in the semantic query expansion field and identifies some future research directions. This article emphasizes that the linguistic approach is the most effective and flexible direction for researchers to follow, while the ontology approach better suits domain-specific search applications. This, in turn, means that development of the ontology field may further open new perspectives for semantic query expansion. Moreover, by employing artificial intelligence (AI) and making most of the query context without relying on user intervention, improvements toward the optimal expanded query can be achieved.

Introduction

The internet has become a significant source of information, and prominent companies in industries such as healthcare, advertising, and trading are using it to promote their services. Furthermore, scholars and researchers use the internet to share their thoughts and experiences. To make the most of this vast amount of information, the semantic web was developed. It aims to connect information semantically. As a result, the number of internet users is rapidly increasing, and people are using search engines and other tools to find information. Because of the diverse cultural and educational backgrounds of internet users, they have different ways of searching for information and use various formats and structures for their queries. Some prefer using long statements, while others use short statements.

The primary link between users and web information is formed by information retrieval systems (IRSs). These systems cater to the needs of users regardless of the query structure. When searching for information, users typically enter their query into the IRS and receive relevant documents in response. With the rapid growth of web information, retrieval systems face the challenge of accommodating diverse queries. They must be flexible yet effective in serving both information seekers and providers. A major challenge arises when users submit short queries, which is increasingly common (Zingla, Chiraz & Slimani, 2016). Short queries expand the search scope, resulting in the retrieval of more irrelevant documents.

Commercial search engines use keyword-based mechanisms to retrieve search results that match the exact terms in the user’s query. However, this approach can be limiting for users who struggle to articulate their needs, resulting in an “intention gap” (Selvaretnam & Belkhatir, 2011). To address this issue, researchers and developers have introduced several techniques, including query enhancement, suggestion, relaxation, and expansion (Ooi et al., 2015). Rather than relying on syntactic matching, these techniques aim to improve the contextual meaning of the user’s query (Malik et al., 2022). Query expansion (QE), for instance, requires adding related keywords to the original query to broaden the search area and obtain more results. For example, a search for “study supplies” could be expanded to include terms like “student”, “students” and “supply”. The expanded terms are integrated into the original query to yield more accurate results.

To produce better query results, IRSs are now incorporating the semantic vision of the query. This includes using semantic QE, which involves adding terms to the query that are semantically related to the original terms. For example, if the original query included terms like “student” and “education,” the expanded query might also include terms like “schoolboy”, “learner”, “provide”, “supply”, “stock” and “equipment”. By adding these related terms, the IRS can generate more relevant search results.

Based on the source used, QE can be classified into three main categories: local, global, and semantic. Local-source QE carries using user feedback and query logs as an expansion source. Global-source QE uses a language model corpus as the primary expansion source. Knowledge-based or semantic expansion uses semantic corpora. Semantic QE involves reformulating the query with different words that have the same meaning to produce more relevant results. Three approaches can be used: linguistic-based, ontology-based, and hybrid (Raza et al., 2019). The linguistic-based approach uses linguistic techniques to extrapolate the semantic relationship between words with the help of linguistic dictionaries. The ontology-based approach uses a semantically constructed tree or ontology that connects different topics based on their meaning. The hybrid approach combines both of these approaches.

Although several reviews have been done regarding different aspects of query expansion, there are considerable gaps in these reviews. For instance, works like Azad & Deepak (2019) and Ooi et al. (2015) either just focus on listing various QE challenges—like word ambiguity and vocabulary mismatch—or review different QE techniques respectively. These works are lacking in providing practical case studies which could show the behaviour of QE in a real-world industrial application, especially for non-English or low-resourced languages. Moreover, fundamental works represented by Azad & Deepak (2019) and Carpineto & Romano (2012) mainly address theoretical approaches without considering deep learning and transformer-based advancements, hence being less relevant in modern artificial intelligence (AI) driven contexts. Additionally, most surveys, including (Raza et al., 2019), cover only applications of QE for text data, neglecting the potential of QE with respect to multimedia or diverse data types with low resources. Additionally, as we observe, most of the previous works were published on 2019 and earlier. Therefore, there is an essential need for providing the new and modern frameworks in the filed.

This article addresses the key gaps in previous QE surveys by providing a timely, all-encompassing review that underlines recent advances, especially in AI-based approaches. Contributions are made both for newcomers, who want an introduction to the field, and for experienced researchers, who want to go deeper in their understanding and to contribute to the development of this field. More concretely, we give an in-depth view on semantic QE by defining core concepts, approaches, and techniques of semantic QE. This work presents a survey of the most recent QE frameworks, ranging from 2015 to 2024. Models covered will involve word-embedding-based and BERT-based approaches. Besides describing the most widely used and emerging techniques, major challenges and practical issues of QE are identified across different domains and datatypes with a special emphasis on low-resource languages. Finally, open issues are brought up and future directions are proposed to drive further research in QE.

Rationale

The rationale for conducting this review on semantic QE is driven by the need to address the challenges and limitations of traditional QE. While the number of search engine users increase, the challenges to meet users’ needs also increase. Semantic QE is a powerful tool for meeting this goal. It provides the search process with an effective enhancement mechanism. However, the literature does not provide a comprehensive overview for researchers in the field. The aim of this work is to guide researchers and developers in the semantic expansion field by providing them with a comprehensive review, including the pipeline and tools. In addition, it reviews and compares recent frameworks to stay up to date with the latest advancements in this field. This work covers the revolution in semantic expansion, which includes simple linguistic techniques up to contextual embedding techniques.

Organization of the study

The remainder of the article is organized as follows. “Survey methodology” presents the survey methodology, while “Related works & research motivation” presents recent works that summarize aspects of QE. “Background” discusses background and refinement techniques. “Impact and Applications of Semantic Query Expansion” presents the usage and impact of using semantic query expansion in different fields. “QE categories” presents the QE categories. “Linguistic-based approaches” discusses some recent frameworks based on the linguistic-based approach. “Ontology-based approach” presents the ontology-based approach, and “Hybrid-based approach” discusses the hybrid approach. The “Discussion” section summarizes the challenges and gives an overview of open issues. Finally, “Conclusion” concludes the article.

Survey methodology

To explore the range of query expansion (QE) techniques and their semantic aspects in information retrieval systems, we conducted a well-planned search for relevant research. Our goal was to capture a variety of QE methods, with a particular focus on those using semantic expansion to enhance comprehension and retrieval precision. We used keywords like “query expansion,” “semantic query expansion,” “semantic expansion,” and “query similarity” in different combinations. These terms were searched within titles and abstracts to ensure we covered the topic thoroughly.

Carrying out this survey, we have considered a number of factors that could introduce biases in comprehensiveness and objectivity in our review. Firstly, selection bias has been managed through the systematic inclusions of studies from wide-ranging sources and publication years, focusing on both foundational and recent advances on QE. Extensive searches of several academic databases were conducted in such a manner that relevant studies were included regardless of the publication venue.

Language bias could further reduce the range of studies reviewed, as much QE research is still published in English. This survey focuses on the core of English language studies, but we have tried to include works related to QE applications in other languages, especially low-resource languages, in order to make it more representative at the global level.

Selection criteria were based on predefined categories to avoid any kind of confirmation bias. Varied perspectives were represented, including works that might hint at some limitations and challenges found within QE. Moreover, comments were requested from field experts in order to have a balanced coverage with regard to theoretical and practical aspects concerning QE.

These biases, hence, are the focus of a more objective and comprehensive survey reflecting a variety of researches and applications related to query expansion.

For our sources, we searched digital repositories like ScienceDirect, IEEE Xplore, SDL, and the ACM Digital Library, resulting in an initial set of 343 articles. To refine this set, we applied specific inclusion and exclusion criteria, as shown in Table 1. Each article was reviewed carefully for its relevance and quality, resulting in a final selection of 34 frameworks.

Table 1 Inclusion and exclusion criteria for selecting semantic query expansion frameworks.

Inclusion criteria	Exclusion criteria	
Papers that focus on query expansion techniques within the context of information retrieval	Papers not addressing query expansion in information retrieval are excluded	
Papers published from 2015 to 2024	Papers published before 2015 are excluded	
Papers that clearly describe the framework for query expansion	Papers that lack clarity in describing the framework for query expansion are excluded	
Papers that contribute to a diverse understanding of data types used in query expansion (e.g., text, structured data, etc.)	–	
Papers that discuss the use of AI techniques (e.g., machine learning, deep learning) in query expansion (this is a positive criterion, but not required for all papers)	Papers that do not incorporate AI techniques and do not meet the other criteria are excluded	
Papers focusing on languages other than Arabic (this is a positive criterion, but not required)	–	
Papers eligible for inclusion include journal articles, conference papers, and book chapters related to query expansion	Papers that are not journal articles, conference papers, or book chapters are excluded	

The screening process involved a close review of titles and abstracts to eliminate articles that didn’t align with the survey’s emphasis on semantic QE. We focused on studies that fit our inclusion criteria and contributed to QE within information retrieval. We prioritized frameworks that covered diverse data types, offered clear descriptions, and, when possible, incorporated AI techniques—though the use of AI was an advantage rather than a requirement. Additionally, we gave preference to studies focused on languages other such as Arabic, though this was not mandatory.

After selecting suitable articles, we conducted a full-text review of each to systematically extract key data. We focused on aspects like methodologies, techniques, and dictionary resources used in QE. This final selection offered a well-rounded view of QE methods, underlying frameworks, and semantic approaches. The extracted information provided a foundation for further analysis and synthesis, helping us clarify the key concepts, steps, and recent progress in semantic QE.

The following research questions were applied in this survey for data extraction and analysis of existing frameworks: RQ1: Which data types and sources are most commonly utilized in query expansion methods?

RQ2: What techniques and methodologies are employed to enhance query expansion, particularly those that incorporate semantic approaches?

RQ3: What are the major challenges and future research directions in the field of semantic query expansion?

Related works and research motivation

It is important to compare and discuss existing works surveying QE research to establish the novelty and significance of the present research.

In their research, Selvaretnam & Belkhatir (2011) explored the challenges of QE, specifically addressing word ambiguity and query-document vocabulary mismatch. To overcome these challenges, the authors suggested studying the linguistic characteristics of the query terms, including their morphological and syntactical features. Additionally, identifying the relationships between query terms can help in understanding the direct and indirect connections between a document and the query, thus mitigating the issue of query-document vocabulary mismatch. The authors approached the QE process analytically, breaking it down into a set of modules and describing the key concepts of each one. While their work may be more beneficial for advanced researchers, it provides valuable insights into the complexities of QE.

Ooi et al. (2015) conducted a study that compared different QE techniques. They provided a comprehensive overview of QE methods, including relevance feedback, language models, and corpus-based models. Additionally, they analyzed these techniques in detail and highlighted their differences. The study also discussed the differences between QE, query refinement, and query suggestion. The authors concluded by outlining the benefits and drawbacks of each technique.

Azad & Deepak (2019) and Carpineto & Romano (2012) both provide a strong representation of theoretical approaches, but they fail to present practical case studies that show the behavior of QE techniques in the real world among industrial applications. Indeed, such applications could provide insights into the practical trade-offs and adaptation challenges when applying these QE techniques to different domains. In addition, their work mainly covered traditional QE techniques, with little attention given to the revolutionary use of deep learning and transformer-based approaches. Moreover, their work focused on applying QE techniques in English, with limited attention to challenges faced in low-resource languages such as Arabic.

Raza et al. (2019) made a significant contribution to the discussion of semantic QE and its importance. They discussed various approaches to semantic QE and the methods used to evaluate QE. The researchers concluded their work by briefly outlining two future directions in the field. The study sheds light on the importance of semantic QE and the different ways it can be approached as well as the need for continued research in this area.

A comparison of the present article with recent existing related works in terms of various QE aspects is presented in Table 2. In our comparison, we looked at several key aspects to understand how previous surveys tackled QE. We considered whether each survey explained the main QE pipeline and the different approaches used, as well as the main challenges and open issues in the field. These criteria are crucial because they help set a foundation for anyone looking to get a clear picture of the QE landscape. We also looked at whether these surveys included real-world case studies in low-resource contexts where semantic QE techniques were applied—this is important because it shows how adaptable and practical these techniques are when data is limited. Additionally, we checked if they covered LR languages with different datatypes, as this highlights how QE techniques can be applied in diverse, real-world scenarios. Another factor was their coverage of languages beyond English, which we feel is critical in today’s global context. Finally, we assessed how much attention they gave to AI advancements in QE, given that AI is now transforming how we think about and approach information retrieval.

Table 2 A brief comparison between our paper and the recent existing related work in terms of different QE aspects.

Ref	Pipeline	Approaches	Challenges	Open issues	Practical frameworks	LR datatype: text, images	Multilingual	AI approches	
Carpineto & Romano (2012)	✓	✓	✓	✓					
Selvaretnam & Belkhatir (2011)		✓	✓	✓					
Ooi et al. (2015)		✓							
Raza et al. (2019)		✓	Partially	Partially					
Azad & Deepak (2019)	✓	✓	✓	✓					
Our paper	✓	✓	✓	✓	✓	✓	✓	✓	

According to the table, previous research on QE has covered various aspects, but only some studies have comprehensively addressed all the essential factors. For instance, in Carpineto & Romano (2012), the authors provided a detailed overview of the QE pipeline and a discussion of different QE approaches, challenges, and open issues. However, they should have considered practical applications or frameworks, and their focus was limited to text-based data without addressing other low-resource (LR) datatypes like images. Similarly, Selvaretnam & Belkhatir (2011) explored QE approaches, challenges, and open issues in depth. Yet, they did not provide a structured QE pipeline, which can be crucial for understanding the end-to-end process of query expansion. Additionally, like Carpineto & Romano (2012), their work was restricted to text-based data, needing insight into how QE could apply to other media types. In contrast, Ooi et al. (2015) focused primarily on QE approaches without delving into other significant aspects such as challenges or open issues. This narrow focus makes it harder to understand the practical limitations or potential areas for improvement within the QE process. Raza et al. (2019) also concentrated on QE approaches. While it partially covered challenges and open issues, it did not provide a holistic view of the QE pipeline or practical applications in real-world settings. Azad & Deepak (2019) was more comprehensive than other studies, addressing the QE pipeline, approaches, challenges, and open issues. However, similar to previous work, it needed more practical frameworks and considered the application of QE in diverse LR datatypes or multilingual contexts.

We realize that previous surveys may not cover some of these newer aspects simply because they were written a few years ago, before many of these advancements. However, our review uniquely fills these gaps, especially by focusing on the urgent need for modern, AI-driven QE techniques that work well in today’s multilingual and diverse data environments. By emphasizing recent developments, we aim to meet the current demands in information retrieval, where AI-powered QE can make a difference. In this way, we hope our review not only builds on past work but also offers something new and relevant to the challenges of today.

Background

Traditionally, when a user passes a query to the IRS, documents with the related terms will be retrieved and ordered based on statistical computations. However, with the rapid expansion of information, which has left us with an endless number of documents, this approach loses its efficiency. Therefore, queries should be considered with different perspectives such as QE, in which the query is reformulated with more related terms.

The QE philosophy has been introduced and examined to improve query results, thereby increasing its effectiveness. The technique relies on expanding query terms by adding additional ones that are related to the original ones, then exploring the results based on the expanded query (He et al., 2016; Beirade, Azzoune & Zegour, 2019; El Ghali & ElQadi, 2017). This can help overcome some vocabulary mismatch and query ambiguity problems. These problems appear because of either polysemy or synonyms (Ooi et al., 2015). Polysemy refers to different definitions of a word such as the word “Python”, which can refer to a snake or a programming language. Synonyms, on the other hand, are different words with a similar meaning such as “words” and “terms”.

Generally, the QE technique is carried out through a pipeline structure as illustrated in Fig. 1. First, in the preprocessing step, the query is filtered to maintain only the useful terms. Then, in the feature extraction step, the terms are transferred into word vectors to prepare them for the expansion step. Consequently, the candidate terms are selected by measuring their similarities with the original query terms. Furthermore, the selected terms are ranked based on their similarity scores. Finally, the query is reformulated based on the top-ranked terms. These steps are detailed in the following.

Figure 1 Query expansion pipeline.

Preprocessing

The first step is the preprocessing of the query. This is done by removing unwanted terms, such as stop words, and keeping only the important ones. For example, if the user searches using the query “What is the capital of Saudi Arabia?”, then the words “is”, “the”, and “of” do not play an important role in the search process; instead, they can affect the efficiency of the search application by adding extra and unneeded work. Preprocessing can be accomplished using many steps based on the application’s needs.

The main preprocessing procedures are tokenization, data cleaning and stemming as shown in Fig. 2.

Figure 2 Preprocessing general steps.

Tokenization

In this step, the text is transformed into tokens, which are useful and meaningful elements. For this, we split the text into words or phrases based on the application’s workflow. Any n-gram tokenization could be considered based on the purpose of the framework. The above example presents uni-gram tokenization.

Data cleaning

After breaking the text into tokens, the unnecessary ones will be eliminated. Unwanted tokens could be stop words, symbols, URLs, extra white spaces, or words in different languages if the application only deals with a specific language and the text is written in more than one language.

Stemming

In this step, all the words are changed to their root word. Using the stem form is much more useful than the entered text since most of the external knowledge sources (i.e., dictionaries, lexicons, thesauruses, and ontology) are structured using the stem form of the words.

Many tools and libraries can be used in the preprocessing phase such as Natural Language Toolkit (NLTK, https://www.nltk.org), Industrial-Strength Natural Language Processing (spaCy, https://spacy.io/) and Gensim (Gensim: Topic modelling for humans, https://radimrehurek.com/gensim/), which can be used for cleaning data and stemming.

Feature extraction

The second step in the QE process is the extraction of the expansion features. This step is the main procedure that differentiates between the QE approaches and can be done in several ways.

After gathering useful words from the query to be expanded, they are transformed into vectors. The main idea of presenting the text as vectors is to determine the weight and dimensions of each word for further comparison. This creates a representation in vector space where spatial proximity indicates similarity and eases the computational processing of natural language. As illustrated in Fig. 3, each word will be transformed into a vector with a direction and a magnitude that represents its meaning. Consequently, identifying and manipulating the linguistic similarity of the words will be possible. To construct a vector, its dimensions (i.e., characteristics or features) need to be identified. For example, if we would like to present a group of children at school, we should first define the features of each child. As is shown in Table 3, we have chosen their math grades, English grades and age as the desired features and dimensions. For instance, the word “Jack” will be represented by the values of its features, which are 5, 10 and 7 in the Table 3. Therefore, “Jack” will be located in the vector space as the point (5, 10, 7). Consequently, each record in the table represents a vector of a child. Now, by using the vectors, we can present the data in the space as a point in the vector space as illustrated in Fig. 4. From the representation, we can conclude that Sara and Suen have similar features since they are close to each other in the vector space. On the other hand, Mike and Noor have different feature since their points are far away from each other in the vector space.

Figure 3 General approach to measure the similarity in texts.

Table 3 Example of features presentation: children features which are their: age, math grades and English grades.

Child name	Age	Math	English	
Jack	5	10	7	
Sara	5	5	8	
Noor	6	9	8	
Mike	7	6	4	
Suen	6	7	7	

Figure 4 The representation of the words in Table 3 in the vector space.

Each child is represented by three features: age, math grades and English grades. From this representation, we can conclude that Sara and Suen have similar features since they are close to each other in the vector space.

For QE, the representation takes a deeper path called word embedding. Word embedding is a method that assigns numerical representations to words, facilitating their quantitative comparison and analysis. This technique converts words, originally represented as strings, into vectors within a vector space utilizing chosen features or dimensions. Usually, each word is depicted as a one-hot vector in word embedding. These vectors are subsequently positioned in a continuous space, bringing together words with similar meanings in closer proximity (Mikolov et al., 2018).

To determine the features, several useful natural language processing (NLP) algorithms can be used. In the following, some of the widely used algorithms are defined (e.g., term frequency-inverse document frequency (TF-IDF), bag of words (BOW), and word embeddings.

TF-IDF: is a statistical model that identifies the importance of each word to a specific document in a collection of documents. In doing so, it computes the weight of each word based on its appearance in the document; however, it is offset by the number of documents that contain the word. The main logic behind TF-IDF is that it gives high attention to words that rarely appear in the corpus (Sharma, Tripathi & Tripathi, 2015).

BOW: is an approach that takes advantage of the occurrences of the most frequently used words in a text, which are collected as a bag of words. It builds the vector based on the presence of the words in that bag. It assigns 1 to a word if it appears in the bag and 0 otherwise (Yan et al., 2020).

Word embeddings: Although previous methods were useful for presenting text, they were lacking the ability to capture the semantic properties of words. To overcome this gap, word embeddings models have emerged. These models have received increased attention from researchers recently due to their efficiency. A word embeddings model works based on the distributional hypothesis of Harris (1954), which states that words that occur in the same contexts tend to have similar meanings. This concept was adopted in leveraging large text to build useful representations of the words. This type of representation is called word embeddings and can be formed in two ways: static or contextualised. Static word embeddings is a single fixed embedding for each word in the whole vocabulary, while contextualised word embeddings generates representations of words based on the context. Word2vec is a great example of static word embeddings. It was introduced by Mikolov et al. as an open-source project in Mikolov et al. (2013a) and Mikolov et al. (2013b). Word2vec is a neural net-trained model with two layers. Without human intervention, Word2vec processes text by converting the words to vectors. It generates a full numerical representation of words by considering the text features. It is based on either predicting the words by using the context (CBOW) or predicting the context by using the words (skip-gram) (Mohamed & Shokry, 2020). As a result, Word2vec gathers vectors related to similar words in a vector space. One of the most widely used pre-trained models is Google’s Word2vec model. It has word vectors for around 3 million words that are trained using 100 billion words (Mohamed & Shokry, 2020). In fact, some frameworks, such as in Li, Wang & Yu (2018), have used the Word2vec mechanism to determine the initial similarities between the entered query and a set of documents. Moreover, the GloVe model (Pennington, Socher & Manning, 2014), which is a model for the unsupervised learning of word representations, used Word2vec for training and outperformed some models on word analogy, word similarity, and named entity recognition tasks. Recently, there has been a growing trend towards utilizing contextualized word embeddings in QE and NLP tasks in general such as in Khader & Ensan (2023) and Wang, Huang & Sheng (2023) and Bhopale & Tiwari (2024). The Bidirectional Encoder Representations from Transformers (BERT) is a popular deep neural network that has been used in the field. The intention behind its design is to pre-train deep bidirectional representations from unlabeled text by simultaneously considering both the left and right context across all layers. Consequently, a pre-trained BERT model can achieve state-of-the-art performance across a diverse array of tasks with just one additional output layer during fine-tuning (Devlin et al., 2018).

Terms selection

In this phase, the candidate terms are selected. The selection is performed based on the relationship between the original query terms and the candidate terms. The similarity between them is the key to finalizing the relevant candidate terms list. To measure the closeness of the terms with the original query terms, a similarity measurement method is used that represents the next phase in dealing with the word vectors as illustrated in Fig. 3. In the ontology-based approach, for example, the depth of the term in the ontology is used as a measurement criterion. In the linguistic structure approach, methods such as Cosine similarity, Jaccard similarity, Euclidean distance, and Latent Semantic Analysis (LSA) can be used. After defining the similarity weight of each candidate term, a threshed is determined to choose the most relevant ones. In the following, we briefly discuss the main similarity measures.

Cosine similarity: Regardless of their size, cosine similarity shows the similarity between two documents based on the cosine angle between their vectors in the vector space. The similarity range is from 1 to −1, where 1 means the documents are very similar, almost identical. The narrower the angle, the more similarity between the documents (Futia et al., 2018).

Jaccard similarity: measures the similarity between two vectors using one feature at a time by taking the intersection of both and dividing it by their union. In other words, as presented in Eq. (1), it is the size of the intersection divided by the size of the union, where A and B are the vector representing the terms.

(1) J(A,B)=|A∩B||A∪B|.

The similarity between the features in both vectors is computed. The intersection here represents the number of times the feature equals 1 in both vectors. The union is the addition of three parts. The first part is the total number of times the first vector value of that feature equals 1 while the other one equals 0. The second part is the total number of times the first vector value of that feature equals 0 while the other one equals 1. The third part is their intersection (Verma, Agarwal & Khan, 2017; Rinartha & Suryasa, 2017).

Euclidean distance: represents the shortest path between two vectors. It is the true straight line distance between two vectors. It measures how close two word vectors are in the vector space (Zhang et al., 2019).

Latent semantic analysis: learns latent topics by performing a matrix decomposition on the document-term matrix using singular value decomposition (SVD). LSA is typically used as a dimension reduction or noise-reducing technique. LSA aims to reduce the dimensions of a document by using the topics that present the underlying information in the document (Dahir & Qadi, 2021). It decomposes the m*m documents/terms matrix, where m represents the number of matrix dimensions, into three matrices. The first one is the m*n singular matrix, which represents the terms/topics matrix that assigns the terms with their topics. The second one is the n*n diagonal matrix, where n represents the number of the matrix dimensions, which represents the topics/topics matrix that determines the importance of the topic. The third one is the n*m singular matrix, which represents the topics/documents matrix that shows the distribution of the topics across the documents. After that, it rearranges the matrices based on the descending order of the topics/topics matrix. Finally, it chooses the topics with the highest importance values (El Ghali & ElQadi, 2017).

Terms ranking

To show search results to users, they must be ordered by how close they are to the user’s original query. Since each of the selected candidate terms has a similarity value, the results will be ordered based on that value. Thus, the terms will be ranked based on how similar they are to the user query. To manage the closeness of the terms to the original query terms, a threshold could be established that is used to define the minimum value of the needed similarity score. The selection of the threshold should not be random. Since it has a direct effect on the results, it should be chosen carefully. Indeed, several experiments could be conducted to determine the best threshold value.

Query reformulation

After manipulating the query and exploring the related terms, the selected ones will be used to reformulate the query by replacing the original query terms with the new expanded ones. The replacement may concern one term at a time or multiple terms, depending on the application. Consequently, the documents will be returned based on the newly expanded queries.

Evaluation

Retrieved documents play the main role in evaluating the QE framework. Generally, the performance of QE frameworks could be measured by examining the portion of retrieved documents that are actually relevant. Recall and precision are the metrics used to test an IRS (Singh & Sharan, 2018), as they measure the desired improvement. The recall metric is used if the tests include searching in a specific database, for instance, Text Retrieval (TREC) datasets (https://trec.nist.gov/). It measures the ability of the framework to retrieve all relevant documents in the database, see Eq. (2). On the other hand, the precision metric measures the relevance among retrieved documents from a limitless database, such as the web, see Eq. (3). Moreover, average precision is used with ranked retrieval results, see Eq. (4). It is considered if more relevant documents are required to be returned earlier (Zhang & Zhang, 2009). MAP is the mean value of the average precision for multiple runs of the framework. Precision can be maximized using precise terms in the expansion process, while using general terms can maximize the recall (Selvaretnam & Belkhatir, 2011).

(2) Recall=TotalnumberofdocumentsretrievedthatarerelevantTotalnumberofrelevantdocumentsinthedatabase

(3) Precision=TotalnumberofdocumentsretrievedthatarerelevantTotalnumberofdocumentsthatareretrieved

(4) Averageprecision=∑nprecisionofthetop−nretrieveddocumentsTotalnumberofrelevantdocuments.

Furthermore, to evaluate the QE framework, a baseline model should be chosen to use its results for comparison. The baseline model could be a state-of-art model, pseudo-relevance feedback model, Rocchio model, or a model without expansion. The latter model runs the framework with and without expansion and compares their results (Raza et al., 2019).

It is worth mentioning that user evaluation could be considered in the evaluation process to seek their satisfaction; however, it can increase the computation time since it can add turn-around time along with additional computation.

Impact and applications of semantic query expansion

Semantic query expansion has wide ramifications in most fields, embedding and enhancing the way information retrieval systems interact with data and users. This enrichment, in relevance and accuracy of search results due to SQE, thus forms the potential to cause improvement in multiple fields, right from Search Engines to Healthcare and beyond. The key areas where SQE is certain to make a big difference include the following: Improve IR systems: IR systems are at the heart of SQE and its contributions toward relevance and preciseness of the results obtained. Semantically expanded queries allow search engines and digital libraries to understand the needs of the user even in cases when the query provided by the user is not well-defined or clear. For instance, the query refinement by SQE in web search engines provides results that are more relevant contextually, hence a better user experience. This bears implications not only for general search engines but also for the niche or specialized search engines used across different domains like law, academia, and e-commerce.

Health care and biomedical research: The SQE will be of immense help in such an information-intense sector as health care, where timely and accurate information is highly valued. Such a facility in medical literature retrieval will expand the search query related to symptoms, diagnoses, treatments, or medications by adding synonyms, related terms, or concepts relevant in context. This would allow researchers and practitioners to locate articles or studies that they might otherwise miss. For example, results from databases such as PubMed can be enhanced using SQE to retrieve a more comprehensive set of relevant studies for healthcare professionals. Also, it will enhance the preciseness of the medical queries in clinical decision support systems, hence contributing to better decision-making.

Multilingual search and cross-lingual applications: Perception capability by SQE through the barriers of different languages is one of the most striking applications of SQE. As the internet becomes more global and sources increasingly diverse, SQE can be of great value in supporting multilingual search. Using cross-lingual semantic expansion, for instance, the SQE may enable users to retrieve relevant information across multiple languages without having to reformulate queries in each target language. This is of great relevance to international business and diplomacy, education, and customer support alike. It can be applied to improve the retrieval of multilingual content, such as in Wikipedia, e-commerce websites, or even social networking sites, so that more people are able to access information available on a global scale.

E-Commerce and customer support: SQE, if applied to an e-commerce context, would refine product results by inferring the intent of the customer query and extending into related products, categories, or features. This “laptop” search can be extended to “laptops with extended battery life” or even ”best gaming laptop,” making sure the results are closer toward what the customer is searching for. In this respect, SQE can also extend state-of-the-art customer care chatbots or virtual assistants with a larger degree of accuracy by allowing query extension with synonyms and related terms, making the interaction very intuitive and fast.

Social media and trend analysis: With social media platforms becoming probably the most significant source of information and opinions, SQE may play an important role in the analysis of trends and sentiment. All this-coupled with the synonym and terms expansion contextually related to the main query-will enhance the possibility of catching emerging trends, monitoring public sentiment, or tracking important discussions related to a particular topic. It is expected that SQE should bring out more relevant tweets in the case of Twitter data analytics by expanding the query into including other forms of expression that capture basically the same idea, thereby giving good insight into public opinions that could be helpful in enabling businesses and governments to make informed decisions.

Personalized user experience: With more and more personal digital assistants coming up-like Siri, Alexa, and Google Assistant-SQE can give further impetus to the intelligence of these systems to understand user queries. Semantic expansion of the queries will let virtual assistants provide more precise answers, keeping in mind the user’s preference, location, and context. This would lead to not only seamless but far more personalized user experiences across applications in e-commerce, media, and daily life.

Education and knowledge management: In the educational sector, the SQE would enhance knowledge retrieval through digital textbooks, online courses, and other educational databases. Advanced results when searching, other than mere keyword matching to concepts, terms, and ideas related to the query at hand, would be afforded to students and researchers. This can also semantically extend the queries of users in knowledge management systems so as to disclose to them relevant documents or articles that they might be interested in.

Query expansion categories

Query expansion (QE) approaches are categorized based on the method used to extract related terms. The main categories are: local-context-based, global-context-based, and knowledge-based (semantic-based), as illustrated in Fig. 5. It is important to clarify the criteria used for these categorizations to avoid confusion.

Figure 5 Query expansion approaches.

Local-source category

This approach relies entirely on user feedback. For previously discovered queries, the feedback on the relevance of retrieved documents plays a key role in choosing related terms. Pseudo-relevance feedback (PRF) is an example of this feedback method, which selects related terms from the top k relevant documents (Zingla, Chiraz & Slimani, 2016; Dahir, El Qadi & Bennis, 2021).

Query logs also fall under the local-context-based approach. This method utilizes the interactions between the user and the IRS, mapping the users’ logs to the new query (El Ghali & ElQadi, 2017). While this approach is simple and effective in some cases, it depends heavily on the user’s past behavior, making it difficult to automate in an integrated system.

Global-source category

This category includes methods that rely on external corpora, such as WordNet (Fellbaum, 1998). Terms are extracted based on the statistical probabilities of terms present in the corpus, which is done by establishing a language model. All retrieved documents are included in the extraction process, with candidate terms selected based on their relationship with the corpus terms (Zingla, Chiraz & Slimani, 2016). While both local-context-based and global-context-based methods can help expand queries, the lack of semantic linkage between terms in these approaches may cause query drifting. Using external corpora increases the generality of the terms, but since it is based on statistical calculations, there is no guarantee that the extracted terms will be meaningfully related to the original query. Additionally, because all retrieved documents are involved, the process can be computationally expensive (El Ghali & ElQadi, 2017).

While this method is effective for expanding a query, the relationship between the terms is typically statistical and context-dependent, rather than semantically meaningful. The lack of semanticity makes the distinction between global-context-based and knowledge-based methods more subtle.

Knowledge-based category (Semantic expansion)

Semantic query expansion (QE) represents a more advanced technique in the field. This approach enhances query reformulation by making the query more “intelligent.” It is a key aspect of the semantic web, where information is interlinked. By structuring the query to better capture its meaning, semantic QE enables a more precise understanding of the query. This category focuses in selecting terms that are semantically related to the original query terms, rather than based on statistical patterns. This makes knowledge-based methods more precise, ensuring that expanded terms maintain the semantic integrity of the query.

This category is typically built on a knowledge structure, which can be either a linguistic structure or an ontology. Consequently, the knowledge-based category can be further divided into three approaches: linguistic structure-based, ontology-based, and hybrid approaches.

Linguistic structure approach

In this approach, predefined sources such as dictionaries, lexicons, and thesauri are used. The linguistic structure approach applies linguistic relationship methods between the query terms and terms within these sources. Key techniques in this approach include POS tagging, stemming, and BOW methods, all of which help to reduce semantic ambiguity in the query terms (Selvaretnam & Belkhatir, 2011). The approach relies on predefined sources to measure the semantic distance between terms, ultimately helping to align the query’s meaning with its expanded terms.

The linguistic structure approach is flexible, computationally light, and relies on available sources. However, it can be difficult to construct a domain-specific specification using this structure.

Below are a few well-known sources used to find candidate terms:

WordNet: WordNet is a trusted and reliable resource for linguistic analysis. It incorporates psycho-linguistic concepts, which bridge linguistic factors and psychological concepts, alongside computational theories. WordNet organizes words into synonym sets and illustrates relationships between them. It includes definitions and examples for each word sense (Fellbaum, 1998).

Wikipedia: Wikipedia (https://en.wikipedia.org/) is a collaborative online encyclopedia available in 285 languages, with over 55 million articles. Each article provides detailed explanations of specific words, subjects, or phrases and includes links to related topics within the platform.

Ontology approach

Ontology represents a formalized structure of lexical terms and is central to the semantic web. Ontologies define relationships between concepts and the meanings of terms within a given domain. Using tools such as the resource description framework (RDF), an ontology constructs a semantic map among related terms. Ontologies are effective in capturing the meaning of terms but can be computationally complex and difficult to integrate into systems.

Ontology-based methods involve careful curation to ensure their accuracy, as they require specific and clear definitions to be effective. SPARQL is often used as a query language for manipulating data stored in RDF format. Examples of well-known ontologies include DBpedia and CYC.

DBpedia (https://wiki.dbpedia.org/): DBpedia is a community-driven project that extracts structured information from Wikipedia and organizes it in a machine-readable format.

CYC (https://www.cyc.com): CYC is an inference engine that contains fundamental human knowledge, including facts, rules of thumb, and reasoning about everyday life events (He et al., 2016).

CRISP (https://bioportal.bioontology.org/ontologies/CRISP): The Computer Retrieval of Information on Scientific Projects (CRISP) is a biomedical ontology that stores data related to research projects conducted across various institutions.

ConceptNet: ConceptNet is an ontology containing 300,000 words and semantic relationships among them, constructed using well-known semantic sources such as DBpedia and OpenCyc (Jain, Seeja & Jindal, 2021).

Hybrid approach

The hybrid approach combines the strengths of both linguistic structure and ontology-based methods to enhance query expansion. In this approach, the first step involves extracting terms from the linguistic structure, followed by exploring the terms within the ontology to find related concepts (such as parent and child terms). While this approach can yield more semantically relevant terms, it is computationally expensive and may still suffer from query drifting.

Table 4 illustrates a brief comparison between these three structures based on their complexity, whether they are computationally expensive or not, and source aspects such as their availability, integration ability and whether they support domain specification.

Table 4 Semantic expansion approaches.

Approach	Computationally complex	Source availability	Source integration	Support domain specific search	
Linguistic structure approach	No	Yes	Easy	No	
Ontology approach	No	Only domain specific	Difficult	Yes	
Hybrid approach	Yes	Only domain specific	Difficult	Yes	

Based on the above semantic expansion approaches, all the frameworks have been categorized. Figure 6 presents a proportional distribution of the frameworks based on the semantic approach they use. In the following three sections, we examine the frameworks based on the approach they use.

Figure 6 Approach-based distribution of the chosen semantic query expansion frameworks.

Linguistic-based approaches

In this type of approach, the extraction of the related terms is based on the linguistic structure of the query. It relies on predefined linguistic resources (i.e., dictionaries, lexicons and thesauruses) such as Wordnet. The relationships between the query terms and their relevant terms are computed using a suitable NLP algorithm. In this section, for the sake of clarity, we introduce linguistic-based frameworks based on the used techniques: linguistic and linguistic-embeddings.

Linguistic models

Lu et al. (2015) proposed a method to help software maintainers perform updates on projects. Their work mainly focuses on expanding the query to improve the search for code. The main source used is WordNet (Fellbaum, 1998). The authors’ basic methodology is to apply the POS with each word in the query. Then, all the resulting tags will be matched with their related synonyms in WordNet (Fellbaum, 1998). Moreover, the methods identifiers are collected to be used in the query matching phase. Then, the results will be chosen based on their similarity with the natural language phrases that exist in these identifiers. The similarity score is calculated based on the total number of common words between the expanded query and the identifier divided by the total number of words on both. The candidate selection criteria depend on how far the tag is from the threshold set by the authors. To test their approach’s effectiveness, they performed two main evaluation stages. First, they compared the result of the expansion method vs the original one. The main comparison was based on the number of relevant and irrelevant suggested terms. The results show an improvement in precision and recall with 40% and 24%, respectively. Second, they compared their approach with the result of Conqure (Hill et al., 2014). Their approach outperforms its precision by 5% and its recall by 8%. The approach is simple, yet it showed improvement in the code query’s effectiveness due to the consideration of methods identifiers. However, the performance of the approach depends indirectly on the quality of the codes. Hence, using a specific domain ontology would be more useful and comprehensive.

Sharma, Tripathi & Tripathi (2015) provided a semantic expansion technique to overcome the delay in searching for patents by using patent abstracts as a query and the International Patent Classification (IPC) as metadata to reduce the search time. Their method is based on three main stages. Firstly, all the relevant abstracts from the IPC are extracted. The extraction is performed by calculating the Term Frequency—Inverse Document Frequency (TF-IDF) feature. Secondly, the expansion phase occurs on two levels: extracting relevant words from WordNet and extracting reliable page titles from Wikipedia. With WordNet, a vector is established. It contains weights ranging between 0.0 and 1.0 for each word. The closer the new word is to the original word, the higher weight it holds. After that, a vector of incoming and outgoing links is calculated using the Page Rank algorithm on Wikipedia. To avoid failure, each page has from 10 up to 100 links, which means there is limited expansion. Finally, the final semantic similarity will be based on the expanded vectors. Its measurement is conducted by using cosine similarity and the Jaccard coefficient. To test their work, the authors performed experiments with and without the suggested expansion. The precision and recall for the former showed better results. The integration of using different resources is a sufficient improvement. Moreover, the usage of IPC can provide more accurate and faster results. However, in the semantic similarity calculation, the Jaccard coefficient method showed poor performance. Also, they did not clarify a threshold to accept the term based on its semantic similarity score.

El Ghali & ElQadi (2017) attempted to eliminate the ambiguity of a query. They took the environment of the query into consideration by building the context around it with the help of the language models (LM) and applying the LSA method. Their work was based on two main phases. The first phase is the query recommendation phase. This step is based on the user’s past searches. By using LM, the past queries are measured and ordered by their correlations with the new query. This method relies on the presence of a term in a query and the presence of a document among the clicked documents. The ordered list of past queries will then be handled by the next phase. The second phase is the LSA method for QE. This phase considers the conceptual meaning of the words. It constructs a word-document matrix to determine the word usage among documents by using singular value decomposition (SVD). SVD assigns a vector to each term. It focuses on the occurrence of the terms in similar documents and queries, even if they never co-occur in the same query. After that, the vectors will be ranked using the similarity score for each query term by using the Cosine similarity measure. To validate their work, the authors used the CISI text database from SRT. They used the original query results as a baseline. Their system improves the precision by 24% and F-measure by 7.76%. The method took advantage of the user’s query log to define the semanticity based on the user’s preferences. However, the log size should be considered since it can affect the time efficiency.

Goslin & Hofmann (2018) used a states-based approach to generate the candidate terms. It is based on providing a sub-state framework where the surviving related candidate terms for the current state will be moved to the next state to extract more related ones. First, it constructs the root query, which is treated as the basic reference throughout the states. After preprocessing of the query, if its length is three tokens, then a connection is made with Wikipedia pages corresponding to it. Otherwise, the N-grams feature is applied to longer queries. For each token, the term frequencies are stored. Then, each term will be passed to six data utilization modules to gather the corresponding weight of the terms. The six data utilization modules are developed linguistic modules. Consequently, a stem query is generated to be passed to the next state. The stem query is selected based on the terms with the best weights from the last step. To prevent query drift, the original query is passed to the next state along with the stem query. The final selected terms are the output of the final state. To test the approach, they conducted a comparison experiment with five existing algorithms. A total of 50 different web topics were used. Their approach had an overall mean average precision of 80%, which outperformed the other algorithms by 27%. Overall, the approach highlights the importance of using states to examine the term importance. Furthermore, query drift was prevented by appending the original query in each state. However, time-complexity-wise, the algorithm is based on revisiting Wikipedia, which could be computationally expensive.

Dao Thi Thuy et al. (2017) proposed a method for retrieving images by using an image as a query. This method relies on the user to determine the semanticity of the images (Dao Thi Thuy et al., 2017). In addition, it differentiates the features by weighting their semantic importance to gather more accurate results. The main idea of the proposed algorithm is to transfer the user image query into a multi-point query. Each query point will be considered as a cluster in the image database for further search. After the user sends their image for a search, the query image will initially be changed to four representations, which are multiple versions of the image with different color layers. Each representation is a cluster. The feature vectors will be extracted for each of them. Then, the distance between multi-point query and the images in the database will be calculated and assigned based on the minimum distance between the image, the feature importance weight and the query points. The feature importance weight is used to increase the semanticity of the image. Indeed, the feature importance is calculated as the inverse of the variance of the feature in each axis in the multi-feature space. This distance is combined with the semantic weight of each query point, which is calculated using the number of related images in each cluster. Based on the user selection, the selected images will be the clusters’ new centroids, and the above algorithm will be repeated. If the user is satisfied, then the algorithm stops; otherwise, the whole procedure will be repeated with the user intervention until they are satisfied. The method was tested with a database with 34 categories with a total of 3,400 images. It produced a higher precision compared to the other four methods. This method takes advantage of the user’s judgment to determine the semanticity. Also, it brings attention to the feature importance and how it can be used to get accurate results. However, when calculating the semantic weight of each query point, the system relies on a traditional content-based retrieval algorithm. Indeed, building the clusters around the centroids lacks any definition of real semanticity between them; this can affect the actual semanticity since the images’ relatedness is not clearly identified.

Nasir, Varlamis & Ishfaq (2019) proposed a framework to improve a method for automatically enhancing the relevant terms by using external information sources. The semantic measurement is calculated for the query and the related documents and for the query and the candidate terms. Firstly, the related documents are extracted based on the probability of the query terms being on them. Then, an automatic relevance feedback method is used to select the most semantically related documents. For the relatedness between the query and the documents, a combination between three measurements is used; these are omiotis, Wikipedia link-based and pointwise mutual information measurements. The first is based on calculating both the semantic compactness and semantic path elaboration of two words using WordNet. The second measurement is based on the relatedness between terms based on Wikipedia articles. Two terms are related when they have been referred by the same Wikipedia articles. The last measurement measures the co-occurrence of two terms in a large document collection by how often two terms occur together. After that, the top-M documents are selected, and their words and the query terms will be measured with the original query by the above measurement techniques to establish the candidate terms. To test the framework, they used three popular datasets. As a baseline, they applied unigram and Okapi methods. They measured the framework performance with precision and recall. The experiment with Omiotis measurement provided the best results for one of the datasets, while the Wikipedia measurement outperformed the other measurements in different datasets. Also, they compared the framework performance with other existing works, and it outperformed them on some datasets. This framework introduced a multilevel semantic measurement; thus, more related documents will be retrieved. In addition, it considers the relatedness between the query terms and the documents before the expansion. However, the threshold of semantic degree for selecting most related terms is not specified.

Riyahi & Sohrabi (2020) proposed a hybrid recommender system to generate recommendations in discussion groups by utilizing information about both the content and the user. The method has three main stages: content-based filtering, collaborative filtering and hybrid filtering. Based on the content of the posts and tags, similar posts will be collected. The tags will be extracted, and a semantic hierarchical structure of them will be built by using WordNet as an expansion source. In the collaborative-based filtering phase, the implicit ratings of all users are obtained to find the users most similar to the active users. The users’ ratings will be based on their behavior such as their comments or their favorite posts. Consequently, the similarity between the active user and the users will be determined by using Cosine similarity measure or Pearson correlation coefficient. As a result, users similar to the active user will be collected. In the content-based filtering phase, the system will generate a recommendation list of the user’s questions from the hierarchical structure of posted question tags. If the question tags do not exist in the hierarchical structure, a search for their synonyms is conducted using WordNet. As a result, posts similar to the active user’s question will be collected. Finally, in the hybrid stage, the most similar posts with the most interactions from similar users will be returned. The method was compared with other recommendation systems, and it achieved an improvement in precision by 35%. This approach is useful for enhancing the quality of discussion group websites. Also, by suggesting similar existing posts, it saves on system resources by minimizing the need for saving posts that already exist. However, using similarity measurement between the new tag and its WordNet synonyms can detect more related tags and thus more related posts.

In the work of Dhokar, Hlaoua & Romdhane (2021), the authors presented a framework for tweet contextualization. It aims to determine the context of a given tweet by retrieving related documents. To do that, they used semantic QE techniques, where they treat the tweet as a single query. The main aim of employing semanticity is to retrieve as many relevant documents as possible to avoid any mis-contextualization. The main purpose of this work is to find an answer to the question “what is this tweet about?” by viewing a summary of information. This summary is constructed from related documents that represent the tweet concept. To collect such documents, a semantic QE approach (SemQEx) is represented. It relies on WordNet as an expansion source. The query (tweet) is annotated via TreeTagger, which is an NLP tool that applies POS on the query terms. Then, candidate terms are selected from WordNet based on their similarities with the query terms. This is accomplished by considering the synsets and their definitions in WordNet. Indeed, the similarity score is calculated based on the occurrence frequency of the query term in the synsets and its definition. Thus, it is considered as a candidate term if the original term appears as a synset or in its definition. To test their work, they applied a dissimilarity score to test the improvement of their work and succeeded in decreasing the score.

Linguistic-embeddings models

Wor2vec

Zhu et al. (2017) introduce a real-time personalized Twitter search method. The main idea of their work is to consider users’ interests, preferences and semantic features. The approach has four steps: feature extraction, feature representation, candidate generation and ranking. In the feature extraction phase, a tweet stream is obtained. After that, the tweets will be preprocessed to get rid of unwanted ones. The tweets will also be filtered to eliminate redundant ones. Then, for each tweet, nouns and verbs will be extracted as semantic features. In addition, the poster’s information, hashtags, URL and number of comments will be extracted as social attributes and will be used to test the tweet quality. For the query, it will be expanded by using the TF-IDF algorithm. In the feature representation phase, every word will be represented as a vector using Word2vec. In the candidate generation phase, two main factors are considered to determine the candidate list: the result of the rule filter of the tweet and its quality. The rule filter is a Boolean logic keyword model that uses a threshold of the TF-IDF value to specify if the tweet is relevant or not. To test the tweet’s quality, social attributes will be used to train the quality model; then, the tweets will be tested by the model. Then, the semantic score and the quality score will be merged for every tweet. After that, in the ranking phase, a threshold adjustment method is used. This method works dynamically based on recent historical data of tweets. Based on that, only the most relevant and qualified tweets are returned. The framework was tested using 10 days of a Twitter sampling stream. It was compared with other similar frameworks. It had the best score for many queries, and its performance is similar to the optimal one. The approach introduced the consideration of a tweet’s quality with respect to its relevance. However, the threshold adjustment method is based on making random assumptions on tweets’ distribution time.

In the work of Singh & Sharan (2018), they focused on the ranking phase of the expansion. Their approach relies on combining multiple techniques in ranking and filtering results. It follows five stages: building the term pool, ranking, semantic filtering, choosing optimal terms by a genetic algorithm and reweighing the terms. First of all, the term pool is constructed by the top retrieved documents based on Okapi-BM25 as a matching function. After that, four different approaches are used for further expansion of the terms from the pool: Kullback-Leibler divergence, co-occurrence, the binary independence model and Robertson selection value. Then, all resulted terms will be ranked using four different techniques: Borda, Condorcet, Reciprocal and SumScore. They are used for rank combining of the terms. Consequently, the ranked terms will be filtered semantically using Word2vec similarity measure with the TRECCDS corpus. With a genetic algorithm, only the optimal combination of the terms will be selected, where each gene represents a candidate term and each chromosome represents a combination of candidate terms. Recall is used as a parameter of the fitness function. Also, crossover and mutation operators are used to generate the offspring. In the final stage, the Rocchio algorithm is used to reweight the final expanded terms. The approach was tested stage by stage with other state-of-the-art benchmarks and showed good improvements in terms of recall and precision.

Fang, Zhang & Yin (2018) proposed an expansion method that uses the advantages of the semantic and sequential information of the words to build a retrieval system for biomedical documents. Their method is based on the semantic sequence dependency model (SSDM). Initially, they trained a domain-specific corpus by using a subset of the MEDLINE database. The Skip-gram model was the language model used to generate the word embedding. When the user enters their query, all the synonyms will be extracted from the corpus. The SSDM will combine the query keywords with the extracted terms for the QE. This is done by replacing the query keywords with their related terms with the maximum replacement of three words; thus, the expanded queries will be the result of all possible combinations of the query and its related terms. During the combination phase, a score is given to each combination by using Cosine similarity measurement. Also, during the training phase, each document from the database is given a score as well. Consequently, for the final results, a summed weight of the query and the document scores is issued and used to rank the retrieved results. To test the results, the authors compared their work with a benchmark model and with the conventional sequence dependency model (SDM) by using a thousand questions proposed by experts. They got a mean average precision with a higher value of 0.024. The authors presented a thoughtful approach by studying the impact of different combinations of the expanded queries. Moreover, since it uses a neural network for word embedding, it has the ability to detect common phrases that could be useful in domain-specific searching. However, the time complexity for long queries should be considered.

ALMarwi, Ghurab & Al-Baltah (2020) presented a genetic approach that focuses on the weighting phase. They used Wordnet as their main source for expansion. All the synonyms collected from Wordnet go through three weighting approaches. Then, they considered choosing the optimal weight by applying particle swarm optimization. The weights were used as positions of particles of the algorithm. The main purpose of applying optimization is to improve the term selection. To test their proposed work, they collected an Arabic corpus and applied multiple experiments to analyze the effectiveness of including and excluding each weights approach. Eventually, they had good recall and precision for their entered queries.

Zhang et al. (2019) presented a framework that improves the search for locations based on a semantic approach. Their framework contains two main phases: the location-semantic relationship measurement and the top-k typical and relevant object selection. The first phase depends on three major measurements. The first measurement is used to find the location similarity between two spatial objects. Euclidean distance is used here by considering the latitude and longitude of the objects. The second measurement considers the semantic relevancy between keywords and between the documents of the object’s text. Alchemy API is used to extract distinct keywords; then, the intra and inter correlation between the keywords is constructed by using the Jaccard coefficient and IDF, respectively. Both correlations are combined to define the keyword coupling relationships for each pair of keywords. After that, cosine similarity measure is used to calculate the semantic similarity between documents based on the computed keyword coupling relationships. The third measurement combines the Word2vec technique and the convolutional neural network. Then, the final semantic relationship between two spatial objects is determined by combining all previous measurements. The second phase defines the top-k relevant objects based on the Gaussian kernel function. To test the approach, many experiments were conducted. They compared it with the IR-Tree-based top-k algorithm by using a dataset from Yelp that contains 50,000 points of interest. Then, they did a comparison for different portions of the datasets with different k values. Their work presented better results in a shorter time. Their approach proposed using multiple similarity measurements. However, the semantic factor could be improved by using a common corpus.

Mohamed & Shokry (2020) developed a framework that focuses on the Holy Quran. The framework returns all related verses based on the searched concept. A Word2vec model is trained on an Arabic corpus based on CBOW. They started by working manually on a Quran dataset and provided each verse with a suitable topic. To produce more useful word vectors, they trained the Word2vec model using an Arabic corpus built using different resources. After that, the entered query will have an average vector along with each topic of the Quran. Then, using cosine similarity, the first related topic is returned. Consequently, the related verses are collected based on the selected topic. To test the framework, they checked the retrieved verses with Islamic experts and got a precision of 91.95%. Their work presents a useful way of creating a vector of the entered query by averaging the vectors of its terms; however, it is lacking in terms selection since only the first candidate term is selected, which may cause a loss of other relevant terms.

Cakir & Gurkan (2023) used the generative adversarial model to expand the user’s query. They introduced a conditional generative adversarial model that took advantage of the semantic information of the query. They obtained such information by training their sequence-to-sequence generator with a query-document corpus. The most important innovation of their work is the use of semantic conditions along with each query and fed it to the generator as inputs. For the discriminator, they utilized a recurrent neural network model to classify the adversarial output from the generator. To construct the condition for the generator, they applied three different strategies: vectors of the most similar documents to the input query, vectors of the most similar words in the documents corpus to the input query, and vectors with TF-IDF weights. The documents similarity conditioning strategy eventually provided the best results compared to similar models. They evaluated their work by using Word Coverage, which is the ratio of the words selected by the generator as expansion terms to the number of words that exist in the corpus.

On the other hand, Rahaman Wahab Sait & Alkhurayyif (2023) built a framework for Arabic QAS that depends on matching the new query and the queries that exist in the dataset. To increase efficiency, they categorized the desired dataset using the multinomial naïve Bayes named entity recognition (MNB NER) classification technique. This way, when the user enters a query, a quick search is applied to the stored queries. In a scenario where the query is not found, they use the Embeddings from Language Models (ElMo) model to apply vectorization on the entered query to find the closest match. In fact, the ELMo model employs bi-directional LSTM to generate vectors for a specific sentence. This indicates multiple vectors for the same terms based on its usage in the sentence. After that, the matching process depends on applying cosine similarity and choosing the best match. To test their work, they used two datasets: ARCD and TyDiQA. Their work outperformed similar frameworks with a precision of 97% and recall of 96%.

BERT

Furthermore, Khader & Ensan (2023) also took the same path in using intelligent models to generate and rank the terms. Their main focus is to provide candidate terms that are useful for searching COVID-19-related topics, specifically. Their framework relies on extracting contextual information by using the BERT model. In addition, to provide semantic insights of the terms, they used UMLSBERT, which provides embeddings for biomedical terminologies. Thus, both BERT and UMLSBERT models were used to generate the candidate terms. Then, to perform the ranking, a train ranking model is used that is based on their performance against PubMed, a biomedical search engine. The later model had different variations based on the evaluation metric, whether it is precision or recall. They used the TREC_COVID dataset to evaluate their framework and achieved a precision of 79%.

Wang, Huang & Sheng (2023) relied on using two BERT models for searching in long text. The first BERT model is an off-line BERT used to obtain representation (i.e.: embeddings) for the whole document. It performs this task by dividing the documents into passages. To save space and time, they implemented a compression layer to reduce the dimensions for each passage, then all passages embeddings were concatenated to represent the whole document. The second BERT model is an online BERT used to obtain query embeddings. Since their framework is serving long-text search, the online BERT is expecting a long query. In the case of a short query, this model will expand the query by predicting related terms. The expansion is based on the contextualized word embeddings used for the training. Furthermore, the same compression layer is used to represent the final embeddings of the query. Finally, cosine similarity is used to compare between the passage embedding and query embeddings. For the final similarity score of the document, they took the maximum score of the query and each passage of the document. They tested their work using the MS MACRO dataset, and their proposed framework showed better performance than the interaction-focused method.

Bhopale & Tiwari (2024) also leveraged the BERT architecture for QE, document retrieval, and query-document ranking. They attempted to reduce the semantic gap between documents and queries by applying contextual representation. Their proposed framework goal is to enhance the IR by utilizing BERT contextual representations of the document and query. To overcome BERT’s limitations in dealing with long documents, they presented a summarization technique. In addition, they applied the phrase embedding technique for QE. To perform the summarization technique, they employed BERTSUM (Liu & Lapata, 2019), which is a BERT-based model used to calculate sentence scores, to apply extractive summarization. This process is applied in an offline mode; thus, it produces the semantic dense vectors of each document in the dataset and stores them. After that, when the user enters the query, a vector is created and compared to the stored document vectors by using cosine similarity. Furthermore, to perform the QE, they used a Word2vec model that generates embedding vectors for different words and phrases. Then, it outputs the list of candidate words or phrases using cosine similarity. They evaluated their work using two datasets: TREC-CDS 2014 and OHSUMED. They found that the phrase embedding enhanced the query with semantically related words from the same dataset. Their model scored p@10 with 56%.

Table 5 presents a summary of the frameworks based on their techniques, sources, datasets, language used and evaluation metrics.

Table 5 Comparison of the linguistic structure approach frameworks.

Ref.	Year	Feature extraction approach	Term similarity measurement	Expansion source	Search dataset	Language	Evaluation metrics	
							Precision	Recall	
Lu et al. (2015).	2015	POS	The similarity score is the total number of common words between the expanded query and the identifier divided by the total number of words.	WordNet	Codes	English	83%	91%	
Sharma, Tripathi & Tripathi (2015)	2015	TF-IDF	Cosine similarity, Jaccard Similarity	WordNet, Wikipedia	IPC	English		94.40%	
Dao Thi Thuy et al. (2017)	2016	K-mean clustering	The ratio of the number of semantically related images in a cluster and the total number of related images of n semantic clusters combined with distance from the image to the query.	Image Database	Image	English	60%		
El Ghali & ElQadi (2017)	2017	LM	SVD, Cosine similarity measure	Query logs, top clicked documents	Text	English	24% (short queries 40.54% (long queries)		
Zhu et al. (2017)	2017	TF-IDF, Word2vec	Based on TF-IDF values.	Wikipedia	Twitter	English	nCG 33.94%		
Goslin & Hofmann (2018)	2017	Term Frequency	Based on the final score, which is the intersection of the Wikipedia links that have the query with the links that have both the query and the candidate term.	Wikipedia	Text	English	80%		
Singh & Sharan (2018)	2017	Word2vec	Word2vec	Dataset	Text	English	23.9%	32.5%	
Fang, Zhang & Yin (2018)	2018	Words embeddings	Cosine similarity.	Biomedical corpus: MEDLINE database	Text	English	33.62%		
Nasir, Varlamis & Ishfaq (2019)	2019	Not mentioned	Omiotis measurement, Wikipedia link-based measurement, Pointwise mutual information measurement	WordNet, Wikipedia	Text	English	87.54%		
Zhang et al. (2019)	2019	Word2vec	Euclidean distance, Jaccard, IDF, Cosine similarity	Dataset	Text	English	55%		
Mohamed & Shokry (2020)	2020	Word2vec: CBOW	Cosine similarity.	Classic Arabic Corpus	Text	Arabic	91.95%		
Riyahi & Sohrabi (2020)	2020	Not mentioned	Cosine similarity, Pearson correlation coefficient	WordNet	Text	English	72.5%		
ALMarwi, Ghurab & Al-Baltah (2020)	2020	Word2vec	Cosine similarity	Pseudo-relevant documents and Wordnet	Text	Arabic	87%		
Dhokar, Hlaoua & Romdhane (2021)	2021	POS	Occurrence frequency	WordNet	Tweets	English	94.59%		
Cakir & Gurkan (2023)	2023	Words embeddings	CBOE, Cosine similarity	Corpus	Text	Turkish	WC9: 75%		
Rahaman Wahab Sait & Alkhurayyif (2023)	2023	ELMO	Cosine similarity	Dataset	Text	Arabic	97%	96%	
Khader & Ensan (2023)	2023	BERT	UMLSBERT	UMLSBERT: biomedical embeddings	Text	English	p@10: 75%		
Wang, Huang & Sheng (2023)	2024	BERT	Cosine similarity	Dataset	Text	English	MAP@10: 36%		
Bhopale & Tiwari (2024)	2024	BERT	Cosine similarity	Dataset	Text	English	P@10: 56%		

Ontology-based approach

An ontology is a powerful structure built based on the semantic relationship between the terms. Since the semantic relevance is already defined in the ontology, some frameworks have used it to expand the queries. In this section, for the sake of clarity, we introduce ontology-based frameworks based on the targeted domain, whether it is general domain or domain-specific.

General domain

Beirade, Azzoune & Zegour (2019) proposed a Quran-based search engine and built a Quranic ontology. The Holy Quran has special characteristics, and words used inside the Quran may have different meanings compared with when they are used outside of it. The authors constructed and built the Quranic ontology with respect to its linguistic and semantic concepts. Relationships have been established between the key concepts in the Quran (e.g., relation of synonym, relation of antonym, relation of hyponymy and relation of similarity). Their search engine works through several steps. After applying lemmatization for each word, all candidate terms are collected from the Quranic ontology (i.e., all child nodes of the term in the ontology will be collected). Then, the system will collect the terms that are semantically related to the original query terms. Two Quranic search engines were used to evaluate their approach: the Alfanous Quran search engine and Corpus Quran. The approach system gives 70% average precision, which is the highest precision among other search engines. Indeed, using recall as an evaluation metric is required since the system uses a limited dataset, the Quran, to search. The approach’s strength is in the terms expansion phase. However, it does not provide a ranking mechanism since it expands the query with various terms, which can result in a massive number of verses.

Furthermore, Al-Smadi et al. (2019) attempted to build a bridge between Arabic questions and DBpedia. They proposed a framework to create SPARQL queries from the entered question. For preprocessing, they performed tokenization, stemming and named entity recognition. The latter is the most important since it could identify the names in the question. To perform the preprocessing steps, they used a Java library called FARASA. After that, they collected the resource identification and labels, DBpedia in their case. Then, they acquired the properties from the question, which connect the subject with the objects. Since DBpedia does not have Arabic labels for most of its represented properties, they performed the properties collection with different approaches: the baseline approach, Wikidata-based approach, and dependency parsing-based approach. The dependency parsing-based approach generated the best results for their framework. After that, they classified the triples to subject-based triple and object-based triple and formulated the SPARQL query based on that. After evaluating their work, they achieved 84% precision.

The work of Jain, Seeja & Jindal (2021) considered working with fuzzy ontology to show its impact on the QE process. To perform this, their framework has three main phases: fuzzy ontology construction, QE using the fuzzy ontology and retrieving the needed information based on the expanded query. First, to build the fuzzy ontology, a dictionary is created by applying text mining on a certain corpus with the help of external ontologies. The main purpose of the text mining is to extract the main concepts on the selected corpus. Then, the ConceptNet ontology is used to define the relationships of the concepts, whether they are synonymous, hierarchical or functional relationships. Then, based on the type of relationship, a semantic weight is assigned for each pair of the concepts to complete the creation of the fuzzy ontology. After that, the fuzzy ontology is used to expand the user’s query. This is performed by selecting the top three related concepts based on their weights. Then, the information retrieval process is performed using the expanded query. To test their work, they constructed a fuzzy ontology using the Solar domain using a database from the UCI Repository. Then, the expanded query is tested throughout the web search engines. Their framework reached 80% precision among Yahoo, Google and Bing.

Domain-specific

In Yan et al. (2017), the authors focused on searching in the ontology itself using RDF queries. They tried to find solutions to queries with too few results or empty results. They calculate the similarity between the query and its candidate by considering the similarity of two triples in the ontology and the IDF score of the candidate triple. They developed a relaxation algorithm to produce relaxed queries and ranked their importance based on their semantic similarity with the original query. Their methodology was based on ontology relaxation of RDF triples. This means dealing with the entailment in RDF. They studied the semantic similarity with two factors: the semantic overlap-ratio and semantic depth. The former represents the number of mutual elements in two RDF triples. The second depends on the ontologies’ hierarchical depths of the concepts. It is measured by the distance between the hierarchical depths. Also, every RDF is weighted by using TF-IDF. The final similarity degree will be computed by using the semantic overlap-ratio, semantic depth and the RDF triple weight. In the relaxation method, the top-k relaxed queries are then chosen to be executed based on the total similarity degree and a defined threshold. They tested the method by using a dataset with 100 K distinct triples. Queries with empty answers were used. They compared their work with other published methods and got the minimal running times. Their algorithm outperforms other algorithms with higher recall; however, based on their precision results, their ranking methodology needs further development.

Deepak & Priyadarshini (2018) proposed a feedback-based approach using appropriate ontologies for homonyms to retrieve images semantically. The first phase of the system focuses on building the ontology. The ontology is constructed for the homonyms of the search keywords. Multiple ontologies can be added as required. The system saves the homonymous list to guide the homonym LookUp Directory, which is a HashMap that has a key for multiple values. These keys are considered as indices to the ontology tree. When the user enters the query, after it gets cleaned, the remaining words will be passed to the semantic algorithm. Based on homonym LookUp content, this algorithm is used to choose the related ontologies. After that, they will be classified using the SVM classifier. After the classification, the semantic similarity is measured between the terms and the class labels of the ontologies. An OntoEntity is then formulated by adding the matching class label, the matching homonym LookUp content and the similarity score. Further, the similarity is measured between the OntoEntity and the metadata (i.e., the description of the image). When the user sees the results and clicks on one of them, this decision will be considered later in the algorithm and used as content information. Thus, the algorithm will work based on the lookup directory along with the content information and construct the semantic similarity by using semantic equivalence matching. The content-based analysis is done by keyword matching between the query original terms and the selected image URL words. Based on that, the classes of ontologies will be prioritized and the most semantically related ones will be displayed. The semantic measurement is done using pointwise mutual information measurement. The proposed framework was tested and compared with four other similar works. It achieved 95.33%, 96.41% and 95.87% on precision, accuracy and F-measure, respectively. The framework constructed a separate phase for building an ontology repository, which makes updating it much easier. It incorporates user intervention in building the ontology, which may be misleading if the user enters inconsistent data into the ontology. It also incorporates user intervention in the semantic measurement but without relying on it to determine the semantic score.

Al-Nazer, Albukhitan & Helmy (2016) worked on utilizing existing semantic web repositories as well as building a bridge between non-English speakers and English resources. This was done by implementing a natural language interface to the English ontologies. This interface will match the entered query with the most relevant semantic web query written in SPARQL, which can be used to get RDF results. The SPARQL query is created based on the expanded queries of the entered query. The authors’ architecture is built in four main stages. First, the entered query will be preprocessed to eliminate any useful words and then get translated to English. After that, in the query mapping phase, the query is classified to create the correct SPARQL query. After that, if any measurement is listed in the query, it will be determined whether it should use the SPARQL query. Next, semantic mapping is performed between the query terms and ontology name entities to identify the ones most relevant to the ontology topics. Finally, based on the relationship between the terms in the ontology, the relationship between the terms will be identified. As a final step, the SPARQL query will be generated based on all the expanded queries terms and be able to be executed to gain the related RDF results. To test their work, they built an integrated ontology from food and health ontologies. They compare the performance of the approach with a manual approach. A total of 389 questions were used. First, their concepts were manually identified. After that, the same experiment was conducted using the proposed approach. Their approach’s precision, recall and F-Measure are 89%, 79% and 84%, respectively. The proposed framework could capture more semanticy if the original terms were expanded before matching them with the entity names’ components. In other words, if the query terms do not match any topic from the entity names, the SPARQL query could be empty.

In Dahir & Qadi (2021), the authors attempted to utilize DBpedia attributes values by choosing the most appropriate ones. To achieve that, they used an integration of topic modeling to find out how topics and attributes are semantically related. Their methodology has two main phases. In the first phase, the relationship between the query and the document is determined by a language model. Then, Bo1 is used to assign the probability distribution to select the top Bo1 expansion terms from the returned documents. In the second phase, for each Bo1 expansion term, DBpedia attributes are selected, whether they are single or multi-valued. After that, latent Dirichlet allocation (LDA) is applied to Bo1 expansion terms from DBpedia by using two attributes to collect the final candidate expansion terms. LDA is used to find the distribution of the topic in documents and the distribution of words of the document to topics. Their experiments showed that LDA can solve the problem of multi-valued attributes of DBpedia, which means increasing the efficiency of the selected terms.

Table 6 presents a summary of the frameworks based on their techniques, sources, datasets, language used and evaluation metrics.

Table 6 Comparison of ontology approach frameworks.

Ref.	Year	Feature extraction approach	Term similarity measurement	Expansion Source	Search dataset	Language	Evaluation metrics	
							Precision	Recall	
Al-Nazer, Albukhitan & Helmy (2016)	2016	Not mentioned	Based on matching the query terms and name entities in the ontology	Ontology	Text	English	89%	79%	
Yan et al. (2017)	2017	TF-IDF	Based on semantic overlap ratio and semantic depth of two triples	Ontology	RDF triples	English		90%	
Deepak & Priyadarshini (2018)	2018	Lemmatization	PMI: Pointwise Mutual Information	Ontology	Images	English	95.33%		
Beirade, Azzoune & Zegour (2019)	2019	Lemmatization	Collect all child nodes of the term in the ontology.	Quranic ontology	Quran verses	Arabic	70%		
Al-Smadi et al. (2019)	2021	FARASA	Baseline, Wikidata-based, dependency parsing-based	DBpedia	Text	Arabic	84%		
Dahir & Qadi (2021)	2021	LDA	A combination of LDA and Bo1 is applied	DBpedia	Text	English	45%	68%	
Jain, Seeja & Jindal (2021)	2021	Text mining	A fuzzy membership weight using ConceptNet	ConceptNet and external ontologies	Text	English	80%		

Hybrid-based approach

In this approach, the framework produces strongly related terms. It explores the linguistic structure of the term and takes advantage of the strong semantic relationship between the terms already built in the ontology. In this section, for the sake of clarity, we introduce hybrid-based frameworks based on the targeted domain, whether it is general domain or domain-specific.

General domain

He et al. (2016) worked on an image retrieval system based mainly on the CYC knowledge base. Their method is constructed through three phases. First, an interface is provided to the user to pass the initial query. Second, in the QE candidates generator, using the CYC knowledge base, all related terms are extracted. This is done by digging further into CYC ontologies to find related concepts and passing them to the QE candidates ranking system as the expansion candidates array. Third, in the QE candidates ranking system, all the array elements’ semantic similarities will be measured. The semantic similarity is calculated based on the WordNet (Fellbaum, 1998) similarity measurements. After that, the most relevant candidates will be returned to the user to expand their options. Then, based on their deeper selection, the final results will emerge. To prove their approach’s efficiency, the authors used two comparison strategies. First, they tested the system with and without their solution. The results showed that the expansion approach had 27% higher precision than the other one. For the second comparison, they compared the results of their approach with the results of Bing, a well-known search engine. The proposed approach produced more effective candidates. The effectiveness of their approach is 6.4% higher than Bing. The proposed model demonstrates a good interaction between the user and the system. It depends on their dynamic feedback rather than their search history. Also, it uses multiple expansion resources to assess the semantic relationship between the query and the candidates. However, it does not consider the image features in the semantic measurement. It focuses only on analyzing the query text and its related concepts.

Zingla, Chiraz & Slimani (2016) attempted to overcome short queries issues on microblog platforms. Their approach is based on extracting terms using text mining. Their methodology has two main phases. In the first stage, all the candidate terms are generated. In this stage, three steps are used. First, unstructured texts of articles from Wikipedia that are related to the query will be selected. The selection is done by using TF-IDF, and then association rules are applied between query terms and expansion terms to choose the candidate terms. The second step focuses only on the definition of the Wikipedia-related articles. The terms are extracted from the first sentence and the first paragraph of the article. In the third step, SPARQL is used to match related terms of the query with the DBpedia using the description of the concept. In the second stage, the candidate terms will be filtered based on a semantic relatedness measure that includes semantic analysis of Wikipedia and the resulting confidence of the association rules. All the terms that have semantic relatedness measures that are greater than a defined threshold will be selected. To evaluate their work, they performed several experiments on Twitter. The experiments showed good performance when the extraction techniques were combined and both filtering techniques were used.

Futia et al. (2018) illustrate a framework to semi-automate semantic graph generation. They structured the system by using a neural network and used SPARQL queries as a training set. SPARQL is extracted by retrieving all the properties that exist in the queries and stored as URIs. Also, the data of these properties will be retrieved. After that, triples from the queries data are extracted. Then, for each concept among the queries, the label of the highest level classes of the concept is extracted. Before the clustering phase, the neural language engine takes the SPARQL triples and transforms them into sentences for training. After that, vector representation is generated for SPARQL variables, which will be clustered based on the close proximity using cosine similarity. At this point, by using the GUI, the user can edit the results of the clustering. Finally, in the semantic graph mapping phase, the median value of Levenshtein distances is used to assign an attribute in the data source with its close cluster. To test the system, the authors compared the precision of their semantic mapping with the semantic mapping generated by domain experts. The ontology used was DBpedia. The result showed a complete overlap between them. The approach introduced the use of AI in the semantic expansion field. However, to make the framework more intelligent and effective, there should be less user intervention involved and linguistic sources such as WordNet could be useful. In addition, Levenshtein distances focus on the spelling of variables; therefore, it does not consider the semantic relationships. Semantic analysis could be performed by stemming the variables in the cluster and then using semantic measurement to accept the new ones.

Later, Song, Huang & Sun (2019) presented a framework that makes a similar contribution as Al-Nazer, Albukhitan & Helmy (2016). They proposed a semantic query graph to construct the information semantically with the help of a QE approach. The main idea behind this is to translate the NL question query to SPARQL. The framework works in four phases. First, in the query dependency parsing, the question structure will be analyzed by using Stanford Parser, an NLP parsing tool. This analysis includes determining the dependency relations of the question words. Then, the dependency parsing tree is generated based on the relationship between the words. After that, the tree structure will go through an optimization process that includes merging words, removing less semantic dependencies and joining dependencies; then, a dependency parsing graph is constructed. Secondly, based on the graph from the previous step, the semantic query graph is constructed with respect to the main entity on the question, the question type and the verb type if the question contains one. After that, using string similarity score, the semantic similarity is calculated by using WordNet to match the semantic query graph with the knowledge base: BDpedia Spotlight. Then, the relation edges in the graph are mapped to predicates and the entity nodes to the attribute value. Finally, the corresponding SPARQL is generated. This is done by traversing the semantic query graph where the edge represents the SPARQL conditional statement. The SPARQL variable will be the variable node, and the resource will be the entity node. Four different datasets were used to test the approach. The results were obtained by using their approach and three other ones. When their approach was used with two datasets, it showed a good performance; however, the approach did not introduce an optimal solution, as it failed in capturing and translating some questions. Using a different knowledge base may produce better performance of the approach since BDpedia stores the resources in a format that may differ from the ones in the question.

Furthermore, Nehar et al. (2022) established a hybrid semantic statistical QE for an Arabic IRS. They relied on two modules: semantic and statistical. The semantic module is based on using the AraBERT model (Antoun, Baly & Hajj, 2021) by preparing both the queries and the documents to follow the AraBERT representation. Afterward, they trained the MLP model as a relevance classifier. To do so, they used the EveTAR dataset to train and test the model. For the training phase, each query had 10 positive samples (i.e., relevant documents) and 10 negative samples (i.e., irrelevant documents). For the testing phase, they used the documents retrieved by the BM25 retrieval model. On the other hand, the statistical module relied on using the PRF approach by selecting the 1,000 top documents from the initial set resulting from BM25 model. Then, they used RM3 to select the 10 top terms. Finally, to perform the expansion, both terms collected from both modules were used. Their framework reached 72% MAP.

Kumar & Sharma (2022) presented a hybrid approach for QE and applied optimization algorithms to obtain the optimal terms for expansion. Their framework employed a topic mapping dataset to use it as an expansion source. They establish a topic map in order to use it later for terms matching. To do so, they used Wikipedia articles and newsgroup datasets. Once the main topics are extracted, Wordnet is used to add related terms to each topic. Therefore, when the user enters a query, the framework tries to find the related terms from the topic map by applying Jaccard similarity. Then, the resulted terms must be passed to the optimization module. To accomplish that, they used two optimization algorithms: spatial bound whale and binary moth flame. The algorithms work in parallel and, finally, produce the optimal set of related terms for the expansion. They achieved 40% MAP@top-3.

Domain-specific

In the work of Malik et al. (2022), a hybrid QE framework was proposed for biomedical field queries. It aimed to overcome the vocabulary mismatch problem, which can occur due to the diverse lexical variants in the biomedical field. They used three main sources to enrich the query: clinical diagnosis information (CDI), Wikipedia and Mayo Clinic. Their framework consists of five stages. In the first stage, the main goal is to extract biomedical concepts. This is performed by using MetaMap annotation, which extracts the desired concepts from the unstructured clinical notes, and each one is assigned by an ID called CUI (Concept Unique Identifier). In the second stage, queries are created from the extracted concepts and sent to the Google Custom Search Engine (GCSE). Then, the GCSE fetches both Wikipedia and Mayo Clinic results and adds them to the collected related results. After that, the original query is expanded further by using the pre-trained word embedding model by collecting candidate terms that are semantically related to the original query via cosine similarity measurement. They use three types: domain-specific, domain-agnostic and hybrid. Furthermore, the resulted candidate terms and CDI are merged and sent to the PubMed corpus, a pre-processed indexed biomedical corpus, to obtain the best combination for the query. Finally, based on the final query, the related biomedical literature is retrieved. Their experiments showed an increase in precision when they used the hybrid word embedding model.

The work of Lechtenberg et al. (2022) represented a query-by-document framework that aims to take the seed document as a seed and construct a query string. This query will then be sent to an API to fetch for related scientific documents. Their methodology presents a QE mechanism with a different order and purpose than the previous works reviewed. First, they used the seed corpus to formulate the targeted search domain. To do so, a list of keywords are extracted from that corpus to determine the characteristic of the domain knowledge. These keywords are chosen based on their TF-IDF weights. Then, to fetch for the related documents, a query is constructed from the list of the extracted keywords. To accomplish this, they used the Monte Carlo (MC) sampling principle, where the query is generated by choosing candidate keywords from a list that have a probability distribution based on their TF-IDF values. After that, the document frequency is calculated based on its appearance in every MC iteration. Indeed, the document frequency will be used to rank the relevance of the resulted documents. In addition, the semantic rank of the results is calculated based on cosine similarity measurement. To test their work, they used two case studies. In the first one, they targeted two different search fields and attached a high recall value, 83.9%. In the second case study, they compared the methodology with different sampling techniques and concluded that MC reached the highest linguistic relevance results.

Table 7 presents a summary of the frameworks based on their techniques, sources, datasets, language used and evaluation metrics.

Table 7 Comparison of hybrid approach frameworks.

Ref.	Year	Feature extraction approach	Term similarity measurement	Expansion source	Search dataset	Language	Evaluation metrics	
							Precision	Recall	
Zingla, Chiraz & Slimani (2016)	2016	TF-IDF	Based on the Cosine similarity and the maximum confidence of the association rule.	Wikipedia, DBpedia	Text	English	26.55%		
He et al. (2016)	2016	Not mentioned	WordNet Similarity Measurement	CYC, WordNet	Images	English	77%		
Futia et al. (2018)	2018	Word2vec	Levenshtein distances, Cosine similarity	DBpedia, RDF triples	Text	English	60%		
Song, Huang & Sun (2019)	2019	Not mentioned	Based on the length of the greatest common subsequence over the length of the word.	DBpedia, WordNet	Text	English	94%	80%	
Malik et al. (2022)	2022	Not mentioned	Cosine similarity	Clinical diagnosis information, Wikipedia, Mayo Clinic	Text	English	P@5: 48%		
Lechtenberg et al. (2022)	2022	TF-IDF	Cosine similarity	domain specific corpus	Text	English		83.9%	
Nehar et al. (2022)	2022	AraBERT	BM25	EveTAR dataset	Text	Arabic	MAP: 72%		
Kumar & Sharma (2022)	2022	Wor2vec	Jaccard	Wikipedia, Newsgroup, WordNet	Text	English	MAP@top-3: 40%		

Discussion

This section takes a closer look at the current landscape of semantic query expansion (QE), particularly the methods that researchers are exploring around similarity measurement, language processing, data sources, and optimization. Drawing on insights from recent studies, summarized in Tables 5–7, we look closely at the strengths and limitations of these methods and highlight areas where further research could have a real impact. By building on our findings from recent studies (summarized in Tables 5–7), we examine both the strengths and limitations of these methods and identify areas where further exploration could make a meaningful impact.

A clear trend is the popularity of the linguistic structure approach for QE, as shown in Fig. 6. Researchers often favor this approach because it effectively captures the nuances of language and offers flexibility through a variety of linguistic techniques. With several existing corpora available, researchers have a good starting point for QE; however, creating specialized corpora is still an option for those working with niche topics. Relying exclusively on linguistic structures, though, can lead to rigid interpretations, especially in informal or unique language contexts. Hybrid approaches that combine linguistic structures with ontology-based frameworks generally cover a wider scope and often provide better accuracy. However, they can be more computationally demanding, which makes it tricky to find the right balance between accuracy and efficiency.

In similarity measurement, methods like cosine similarity, Word2vec, and TF-IDF are widely used for their ability to capture semantic relationships. Cosine similarity, for example, effectively measures how closely related two vectors are within a topic but struggles with words that have multiple meanings or specific contexts (Mohamed & Shokry, 2020). Word2vec, on the other hand, uses a neural network-based model that captures word relationships more subtly, making it great for processing large text datasets. However, it demands significant computational resources and doesn’t perform as well with languages that have fewer data resources. TF-IDF is useful for stressing key terms in specific documents. However, it can sometimes miss deeper semantic links between words. Given these strengths and limitations, combining TF-IDF with other methods in a hybrid approach may offer a more accurate and nuanced solution for QE.

Our analysis also identified some broader issues within QE, summarized in Table 9. The lack adoption of machine learning and of multi-language usage lead the attention to important gaps. Machine learning shows a lot of potential for NLP and could greatly enhance techniques for measuring similarity, word representation, and context expansion. Additionally, most QE frameworks rely heavily on English resources, highlighting a divide in the field. Very few frameworks are designed for languages beyond English, especially Arabic, which limits their reach and usefulness in diverse regions. Many frameworks are also domain-specific, such as those focused on the Quran, e.g., Mohamed & Shokry (2020), or food-related data, e.g., Al-Nazer, Albukhitan & Helmy (2016). Although these frameworks work well within specific areas, they lack for broader applications. Therefore, there is a need for more flexible, free-domain frameworks. Language diversity poses an extra challenge for non-English users, as most QE frameworks are built with English in mind. This limits accessibility and makes using English-based ontologies more complicated for non-English speakers. A potential solution is using machine translation to convert queries from other languages, like Arabic, to SPARQL queries. Studies like Al-Nazer, Albukhitan & Helmy (2016), Al-Smadi et al. (2019) have proposed translation models, and recent research on Arabic Word2vec (e.g., Soliman, Eissa & El-Beltagy, 2017) shows promising advances in language-specific word representation. These approaches can help filling the gap between English and non-English resources, which will QE to be more accessible to a larger users. However, achieving accurate translations is still a challenge, as even small errors can lead to “semantic drift,” where the meaning of the query shifts subtly.

For QE optimization, a few studies—including ALMarwi, Ghurab & Al-Baltah (2020), Singh & Sharan (2018), and Kumar & Sharma (2022)—explore innovative methods to improve QE results. For instance, particle swarm optimization, as used in ALMarwi, Ghurab & Al-Baltah (2020), helps identify optimal weights for candidate terms, while genetic optimization in Singh & Sharan (2018) selects the best term combinations. While these methods can help reduce query drift and improve QE accuracy, they are resource-heavy and challenging to implement in real-time systems. The use of multiple optimization algorithms in parallel, as seen in Kumar & Sharma (2022), holds promise but may not be practical for all applications due to its complexity and high resource demands. Engaging AI optimization with tools, such as cosine similarity and TF-IDF as fitness scores, could make QE techniques more intelligent and afflictive.

Moreover, when evaluating the current techniques for semantic query expansion, some limitations and tradeoffs arise. Hybrid techniques that combine linguistic structures and ontology based frameworks raise some issues of time complexity and scalability. They might increase the accuracy of the query response, but they are too costly in terms of computational resources making them difficult to implement on huge datasets or in real time systems. Another issue becomes one of threshold selection, in particular applying thresholds with similarity measures based on cosine similarity, Word2vec and TFIDF. Ensuring appropriate thresholds for including or excluding terms is important to ensuring the expanded query is useful. Inseting a threshold which is too low tends to be much more undesirable since it is most likely that it will negate the completeness of the expanded text.

Furthermore, the user context as well as user intervention is desirable when dealing with machine learning or any translation models that demand a translation suited to a language and intent of the user. Query drift is another problem which is compounded in the case of machine translation or optimization. The problem arises because even small inaccuracies can transfer the general meaning of a query term, hence changing the accuracy of a query. Lastly, there are considerable challenges, particularly with regard to security, especially concerning sensitive data and application QE techniques in domains where privacy is a significant concern. Such issues pose critical gaps of improvement in a bid to provide more effective, adaptive and secure query expansion approaches.

In summary, while current QE methods provide a strong foundation, they have clear limitations, which opens the door for opportunities for future research to develop. Future work that advances hybrid linguistic-ontology models, expands multi-language support, and integrates AI-driven optimization could help create more flexible and inclusive QE frameworks, making them useful for a larger area of applications.

Challenges

Based on an overview of recent works on semantic QE, summarized in Tables 8, 10, 11, various challenges and open issues can be identified and explored. These challenges and issues must be addressed as they may significantly impact the system’s performance. It is imperative to carefully analyze and address these challenges to ensure the optimal performance of the system.

Table 8 Linguistic structure frameworks summary.

Ref.	Search domain	Key concept	Advantages	Limitations	
Lu et al. (2015)	Codes	Matching exact POSs pairs of query terms and candidates.

Extracting methods identifiers in the code to match them with the candidates.

	Taking advantage of the functions identifiers in the expansion process.	The approach performance based on the code’s quality. Ontology will perform better.	
Sharma, Tripathi & Tripathi (2015)	Patents	Using the patent abstract as query and International Patent Classification (IPC) as metadata to reduce the search time.	Integrated sources, the use of IPC to get more accurate and fast results.	They did not clarify a threshold to accept the term based on its semantic similarity.	
Zhu et al. (2017)	Tweets	Consideration of the user’s semantic and social features.

Defining a boolean logic keyword relevance filter.

Defining tweet quality model

	Tweet’s quality consideration.	The threshold adjustment method is making random assumption in the tweets’ distribution time.	
El Ghali & ElQadi (2017)	General	Extracting query context using query logs.

Applying the LSA method using the query context.

	Using the user’s query log to define the semanticity based on the user’s preferences.	The log size could affect the time efficiency.	
Nasir, Varlamis & Ishfaq (2019)	General	Applying different levels of semantic relatedness; a document against a query and each query term with terms in the document.	Providing multiple measurement methods.	The threshold for the semantic degree is not specified.	
Dao Thi Thuy et al. (2017)	General	Using the user’s selected images to be the centroids of the clusters.	The consideration of the variance of the feature to determine its importance.	The phase of building the clusters around the centroids lacks indefining the semanticity between them.

The main methods are centered around the user intervention, which could be misleading.

	
Mohamed & Shokry (2020)	Islamic holy book: Quran	Building and training a Word2vec model using CBOW on the Arabic corpus and using it for Quran searching.	The construction of a vector for the query as a whole by averaging the vectors of its words.

Applying the machine learning technique to construct the vectors.

	Only the first candidate term is selected, which may cause the loss of other relevant terms.	
Goslin & Hofmann (2018)	General	Providing n sub-state framework where the surviving related candidate terms for the current state will be moved to the next state to extract more related ones.	The persistence of the original query across states to minimize query drift for generated enhancement terms	The algorithm is based on revisiting Wikipedia, which could be computationally expensive.	
Fang, Zhang & Yin (2018)	Biomedical Articles	Integrating both sequential and semantic information to expand the query. Investigating which word-embedding algorithm can best serve for biomedical articles.	Ability to detect common phrases, which could be very useful in domain-specific searching.	The time complexity for long queries should be considered.	
Riyahi & Sohrabi (2020)	Discussion groups	Applying semantic measurement in a recommender system for discussion groups using both content and user information.	By suggesting similar existing posts, it saves the system resources by minimizing the need for saving posts that already exist.	Similarity measurement between the new tag and its WordNet synonyms can detect more related tags and thus more related posts.	
Zhang et al. (2019)	Locations	Using semantic query expansion in spatial keyword query.	The consideration of relatedness of among locations and among their textual documents.	The system precision is low.	
Singh & Sharan (2018)	General	Using different similarity measurements to expand the query semantically and using a genetic algorithm to select the optimal candidates.	Multiple ranking and filtering	The initial term pool should be generated out of semantically related documents.	
ALMarwi, Ghurab & Al-Baltah (2020)	General	Applying several similarity measurements and applying optimization to select the optimal weight for each candidate	Multiple ranking and filtering	The initial term pool is selected from Arabic Wordnet, which is a limited source	
Dhokar, Hlaoua & Romdhane (2021)	Tweets	Reling on semantic query expansion for tweets classification	Multiple ranking and filtering	The initial term pool is selected from Arabic Wordnet, which is a limited source	
Cakir & Gurkan (2023)	Text	Using conditional GAN model to generate the candidate terms with different condition strategies	Increasing the similarity measurement quality by adding similarity condition along with the original query as an input to the model	The huge capacity needed to store the lookup table to match each query with its condition	
Rahaman Wahab Sait & Alkhurayyif (2023)	General	Using stored queries for matching	Elmo vectorization	They used query matching instead of expansion which may cause query drifting.	
Khader & Ensan (2023)	COVID-19 information	Using BERT and UMLSBERT model for expansion	Taking into consideration both contextual and semantic information	Time complexity could be an issue sine they use ML for terms extraction and ranking	
Wang, Huang & Sheng (2023)	General	Applying representation-focused framework for searching in long text	The usage of off-line BERT training to overcome the time complexity	The model semantic efficiency depends completely on the dataset used for training	
Bhopale & Tiwari (2024)	General	Applying sentence BERT embedding of document summary	The usage phrase embedding for query expansion	The phrase expansion is aligned with the used dataset which may failed for words that do not appear in the dataset	

Table 9 General comparison of the literature.

Ref.	Machine learning	Support none English	Linguistic approaches	Ontology approach	Hybrid approach	Optimization	NL1 to SPARQL	Domain specific	
Lu et al. (2015)			✓					✓	
Sharma, Tripathi & Tripathi (2015)			✓					✓	
Dao Thi Thuy et al. (2017)			✓						
Dhokar, Hlaoua & Romdhane (2021)			✓						
Zhu et al. (2017)			✓					✓	
El Ghali & ElQadi (2017)			✓						
Nasir, Varlamis & Ishfaq (2019)			✓						
ALMarwi, Ghurab & Al-Baltah (2020)	✓	✓	✓			✓			
Mohamed & Shokry (2020)		✓	✓					✓	
Goslin & Hofmann (2018)	✓		✓						
Fang, Zhang & Yin (2018)			✓						
Riyahi & Sohrabi (2020)			✓						
Zhang et al. (2019)	✓		✓						
Singh & Sharan (2018)	✓		✓			✓			
Cakir & Gurkan (2023)	✓	✓	✓						
Rahaman Wahab Sait & Alkhurayyif (2023)	✓	✓	✓						
Khader & Ensan (2023)	✓		✓					✓	
Wang, Huang & Sheng (2023)	✓		✓						
Bhopale & Tiwari (2024)	✓		✓						
Dahir & Qadi (2021)				✓					
Al-Smadi et al. (2019)		✓		✓			✓		
Beirade, Azzoune & Zegour (2019)		✓		✓				✓	
Deepak & Priyadarshini (2018)				✓					
Yan et al. (2017)				✓					
Al-Nazer, Albukhitan & Helmy (2016)		✓		✓			✓	✓	
Jain, Seeja & Jindal (2021)				✓				✓	
Zingla, Chiraz & Slimani (2016)					✓				
He et al. (2016)					✓				
Futia et al. (2018)	✓				✓				
Song, Huang & Sun (2019)					✓		✓		
Malik et al. (2022)	✓				✓			✓	
Lechtenberg et al. (2022)					✓			✓	
Nehar et al. (2022)	✓	✓			✓				
Kumar & Sharma (2022)	✓				✓	✓			
Note:

1 Natural language.

Table 10 Ontology frameworks summary.

Ref.	Search domain	Key concept	Advantages	Limitations	
Beirade, Azzoune & Zegour (2019)	General	Building a Quranic ontology based on the semantic relationship between the concept and its child nodes.	Using the topic’s nodes as candidate terms.	It lacks in providing a useful ranking method.	
Deepak & Priyadarshini (2018)	General	Constructing a feedback-based approach using appropriate ontologies for homonyms to retrieve images semantically.	The ability of building an up-to-date ontology.

Taking the user’s click as a useful intervention without relying on it.

	The ontology construction involves user intervention, which may cause inconsistent ontology data.	
Yan et al. (2017).	General	Calculate the similarity between the query and its candidate by considering the similarity of two triples in the ontology and the IDF score of the candidate triple.	The consideration of triples similarity.

The consideration of the similarity between the original query and the expanded query to determine the top-k relevant ones.

	Based on their precision results, their ranking methodology needs more development to capture more relevant results.	
Al-Nazer, Albukhitan & Helmy (2016)	Health /Food	Addressing the translation of the natural language query to the SPARQL semantic query using multilingual and cross-domain ontologies.	It supports multilingual search.	It lacks in expanding the query terms before matching them with the ontology topics; if the query terms do not match any topic from entity names, the SPARQL query could be empty.	
Dahir & Qadi (2021)	General	Combining Bo1 and LDA to determine the best DBpedia attributes to fetch the suitable expansion terms	It showed the effect of different DBpedia features.	Some suitable candidate terms may not appear due to the fixed number of the Bo1 candidate terms	
Jain, Seeja & Jindal (2021)	Specific	Constructing a fuzzy ontology using external ontologies for a specific domain and use ConceptNet to determine the weights.	It presents an integration of different ontologies which could serve on the source availability challenge.	Its weights measurement depends on other sources weights.	
Al-Smadi et al. (2019)	SPARQL queries	Constructing a SPARQL query based on query matching	It can build a brigade between Arabic NLP and SPARQL	It lacks in providing condition matching between the query and SPARQL query	

Table 11 Hybrid frameworks summary.

Ref.	Search domain	Key concept	Advantages	Limitations	
Zingla, Chiraz & Slimani (2016)	Microblogs	Applying a text mining-based approach along with external knowledge sources to expand short queries	The use of association rules results in more accurate related terms	It consumes time since it includes visiting two external sources.	
He et al. (2016)	General	The usage of CYC as a semantic expansion source with the help of WordNet to find more related images.	The use of multiple expansion resources to assess the semantic relationship between the query and the candidates.	It does not consider the image features in the semantic measurement. It focuses only on analyzing the query text and its related concepts.	
Futia et al. (2018)	General	Training a neural language model using SPARQL to reconstruct semantic mapping of a data source.	Providing a semi-automated approach that can be used to build an ontology.	Levenshtein distances focus on the spelling of the variables rather than semantic distance.	
Song, Huang & Sun (2019)	General	Applying query expansion in the natural question to translate them to SPARQL queries.	It can be useful for end users with less experience in constructing SPARQL queries.	The closeness measurement should be considered more since it does not always produce the optimal answer.	
Malik et al. (2022)	Biomedical	Combining external expansion sources with word embedding model	The usage of multiple sources decreases the vocabulary mismatch	The relationship between each term and the whole query should considered since the query represents a particular biomedical case	
Lechtenberg et al. (2022)	General	The usage of Monte Carlo sampling principle to reformulate the query	It can perform well with less seed papers as corpus	The speed of the sampling process is a challenge. Also, it relies on the targeted API limits of presented results.	
Nehar et al. (2022)	General	The usage of AraBERT model as semantic embedding and PRF as statistical source.	Taking advantage of the pre-trained transformer for Arabic	The MLP model needs to be train with more data to achieve better performance.	
Kumar & Sharma (2022)	General	The usage of different source to create topic mapping and two optimization algorithms.	Reducing the redundant terms by selecting optimal ones with respect to their semantic features	The topic model lacks in provide related terms to the words that are not noun or verb	

Source availability

The use of existing knowledge structures, such as WordNet, can be limited when it comes to incorporating newly adapted slang phrases. To address this issue, a utility that can match slang phrases with their exact meaning can be helpful. However, the limited number of existing ontologies poses another challenge. Building a new ontology requires time and comprehensive knowledge of the ontology domain, which can be a difficult task. Despite this, researchers can collaborate with domain experts to develop a comprehensive and efficient ontology. Additionally, most researchers use domain-specific ontologies, but integrating multiple ontologies can increase usage and make ontologies more generic.

Developing a semantic framework for languages other than English is a challenge. To overcome this, a hybrid framework that uses different weighting techniques can be used to determine the similarity scores of candidate terms, as demonstrated in the Arabic language framework presented in ALMarwi, Ghurab & Al-Baltah (2020). A translation phase is also necessary to take advantage of the available English sources; however, the translation must be precise to generate accurate terms and prevent query drifting. To address this challenge, El bazi & Laachfoubi (2015) proposed a framework that can recognize named entities for Arabic texts using the MADAMIRA tool to capture Arabic features and avoid any loss of important words in the translation process.

User intervention

The success of an approach can be influenced by user intervention. Depending on the application, the user’s intervention can explicitly shape the semantic vision of the framework, which may lead to misleading results. For instance, in Dao Thi Thuy et al. (2017), the algorithm’s desired clusters are determined by the user’s selection of related images. However, there is a possibility that the user’s perception of the image characteristics may differ from how they are presented in the image dataset’s vector space. This could lead to query drift and affect the approach’s performance. Therefore, it is crucial to consider the necessary level of user intervention in the framework without impeding the expansion process (Raza et al., 2019).

User context

Understanding the user’s tendencies is crucial in providing them with relevant information, and social and personal information can be a factor that helps achieve this goal. For instance, if a user’s personal information indicates an interest in computers, a search for “Python” should yield results related to the programming language rather than the reptile. However, it’s important to carefully consider the need for user context. While social applications like Twitter can benefit from user context (Zhu et al., 2017), it may be useless and time-consuming on other platforms. Therefore, it is essential to analyze user context and gather it only if it leads to more desirable results.

Time complexity

In the field of semantic QE, the more complex the computations, the more time they consume. In their article, Goslin & Hofmann (2018) proposed a framework that goes through different states, each with its own set of complex computations, making it computationally expensive. Additionally, in Li, Wang & Yu (2018), multiple types of similarity crosschecking were used, including constructing a concept tree and using average mutual information to calculate the similarity between words, which can be time-consuming and impact overall performance.

Therefore, it is crucial to consider time complexity when developing a framework. A reasonable balance must be found between providing accurate results and the time it takes to do so. Users generally expect their results in a timely manner, making this a significant factor in assessing a search application. To improve time complexity over a wider range, solutions such as parallel computing and HPC can be considered. However, questions may arise, such as how to use these solutions effectively, which computations to split in a parallel manner, and whether these solutions will deliver accurate results with less time consumed.

For example, in Kim, Bhattacharyya & Anyanwu (2019), the researchers suggested data and query transformations to execute SPARQL queries in parallel. Their main idea is to reduce the number of join operators and increase the number of union operators for the fragmentary steps of the query, allowing for inter-operator parallelism of ontological queries. This enables parallel execution across all partitions.

Threshold selection

In frameworks that rely on a specific threshold to select the final expanded terms, the accuracy of the threshold selection can significantly affect the framework’s overall performance. Choosing a large threshold may result in losing important and relevant results, while selecting a small threshold may generate numerous irrelevant results. An adaptive threshold, as defined and used by Zhu et al. (2017), can enhance the flexibility and feasibility of the searching framework. By conducting a historical study or data analysis or by considering user social behavior, an adaptive threshold can be effectively structured and implemented.

Scalability

To cater to the diverse needs of users in a search application, the system’s design should be versatile enough to work with various types of databases. While TREC is a reliable dataset commonly used in this field, solely relying on it may not ensure optimal system performance. Therefore, it is crucial to test the scalability of the system, particularly if it is intended for general-domain searching. By doing so, the system’s adaptability and feasibility can be enhanced.

Query drift

QE is a valuable technique that can improve the relevance of search results; however, it is not without its challenges, including the risk of query drifting. Query drifting occurs when the expansion process is too broad, causing the search to produce unrelated results. This can be due to issues with term extraction, biased term weighting, or biased retrieval models (Al-Shboul & Myaeng, 2014). Low precision and recall are indicators of potential drifting, making it a significant challenge in IR model design. One solution is to set a suitable threshold that can generate more relevant groups of expanded terms. Additionally, tracking the original query, as demonstrated in Goslin & Hofmann (2018), can help narrow the focus of the expanded terms to meet user needs. In more advanced frameworks, optimization techniques such as genetic algorithms (Singh & Sharan, 2018) and particle swarm optimization (ALMarwi, Ghurab & Al-Baltah, 2020) have been used to select the most relevant terms, thus reducing the risk of drifting. It is critical to consider these challenges carefully when designing an effective IR model.

Security

Frameworks that leverage user feedback for better results are often based on the analysis of user search logs (Azad & Deepak, 2019). This approach can effectively detect users’ interests and improve the relevance of results. However, it is important to consider the potential privacy risks associated with such logs, as they can reveal sensitive user data (Carpineto & Romano, 2015). To ensure the trustworthiness of retrieval systems, user privacy must be a key consideration. For example, in Bater et al. (2020), the authors proposed a framework that uses secure computations to prevent data leaks from the provider’s side.

In addition, protecting the privacy of queried data is equally crucial. User privileges should be clearly defined to prevent unauthorized data access or modifications. To mitigate such risks, the authors of Hosseinzadeh Kassani, Schneider & Deters (2020) proposed a secure framework based on a blockchain structure. Their approach involves using blocks to store user transaction information and access roles, similar to the blockchain. The hash mechanism is used to securely store data and prevent unauthorized access.

These challenges are highly interconnected with several open issues in semantic query expansion. For example, limited multilingual resource availability, e.g., for Arabic, requires developing multilingual ontologies or lexicons to help bridge the language gaps. Again, integrating AI into QE approaches raises a problem of scalability and complexity due to time complexity. This could suggest solutions which can make query expansion more adaptive and efficient using AI-driven methods. On the other hand, the increasing need for real-time data access and semantic search over multimedia content requires effective frameworks. In fact, such frameworks should be able to deal with dynamic and disparate types. This could increase scalability and effectiveness of QE for different applications.

In the following section, we elaborate deeply in the possible open issues and future research directions.

Open issues and future research directions

The revolution of semantic QE has enriched the field of semantic web and IRSs in various ways; however, there are still open issues that should be the focus of future research directions and improvements. Constructing an ontology can be a time-consuming and demanding task. Therefore, it is important to develop an algorithm that can automate this process while integrating the expertise of ontology developers and experts. For instance, in Deepak & Priyadarshini (2018), an approach is proposed that accomplishes this process in a more controlled and careful manner.

The availability of ontologies and lexicons is limited in languages other than English, such as Arabic, which can hinder the expansion of candidate queries. Building a multilingual ontology or lexicon can improve the QE to overcome this limitation. In Al-Smadi et al. (2019), an automated question/answer system is introduced that bridges the gap between Arabic questions and linked data by translating natural language questions to SPARQL queries to retrieve answers from linked data, such as Dbpedia. To achieve this, the named entity of the original query is discovered with the help of Wikipedia labels available in Arabic. In contrast, AlAgha (2015) utilizes the statistical parser of the Arabic Toolkit Service to identify the subject, object, and predicate of the Arabic text and construct the corresponding RDF triple. To improve the reliability of linguistic resources, slang phrases must also be considered. N-gram and word/sentence embedding can play a significant role in query processing, aiding in catching such phrases with the help of reliable sources.

Applying AI to the expansion field is a challenge, but it could add intelligence to the approaches and generate accurate and useful expanded queries. Deep learning, for instance, could help in recognizing hidden patterns among texts or images, which can provide more semanticity. Furthermore, it can provide contextual analysis of the text and match it with an accurate topic; thus, the framework can proceed to fetch the closest topic. For instance, Mohamed & Shokry (2020) represents a framework that applies a machine learning approach that aims to train a Word2vec model using CBOW on an Arabic corpus and use it for searching the Quran. It scored a high precision by increasing the intelligence of the vector structure. In addition, Fang, Zhang & Yin (2018) used a trained Word2vec model to capture both sequential and semantic information of biomedical texts. Furthermore, in applications that focus on searching in domain-specific areas, machine learning will be useful since the training dataset is precise and determined. For instance, in Zhong et al. (2020) the authors used a deep learning methodology to construct an answer/question system for building regulations to help engineers in retrieving needed information. In addition, Cakir & Gurkan (2023) and Khader & Ensan (2023) showed a great potential in using generative models that can expand the query in an intelligent way.

In the realm of semantic QE, incorporating real-time data is an area that is yet to be fully explored. However, a real-time semantic search has the potential to provide valuable insights on trending topics, especially during emergencies and major events, when people tend to scour social media platforms for information. By leveraging the user’s semantic attributes, a framework described in Zhu et al. (2017) was able to effectively search Twitter stream data, yielding more accurate and interesting results.

The semantic search for multimedia content, such as images and videos, is still in its nascent stages. Existing frameworks rely on user intervention to determine semanticity, as seen in Dao Thi Thuy et al. (2017), which can be misleading and fails to provide an actual semantic expansion approach. Incorporating deep semanticity with minimal user judgment can lead to better semantic frameworks. One way to achieve this is by constructing a vector space of the images based on multiple features to create a digital description of the multimedia content. This approach was used in Tautkute & Trzcinski (2021), where the authors aim to extract semantic information from different data forms for image searching. They use a GAN framework to generate a synthetic-related image based on a multimodal query containing both an input image and textual description, which is then used to search for more related images.

Conclusion

It can be considered that the internet, in this digital era, is a giant storage unit of many forms of information that serve to help several industries and give a platform for various researchers and professionals to publish their views. However, the vast amount and variety in web data turn the job of extracting specific information really arduous, especially because users’ queries reach the search engines with variant structures, level of detail, and expectations. That is where query expansion, especially semantic QE, plays a revolutionary role. Semantic QE develops the conventional retrieval techniques with related terms using terms that exactly capture the intended meaning of the users’ queries, filling the gap between the user’s needs and the responses of the retrieval system.

The article has given an overview of QE, emphasizing recent semantic QE techniques from 2015 to 2024. We tried to illustrate how such approaches, which we grouped into linguistic-based, ontology-based, and hybrid approaches, can be utilized to enhance retrieval precision for various domains. In this review, we emphasized the incorporation of modern AI approaches such as word embeddings and BERT models, which are really transforming the way QE works. We also emphasize the challenges that are presently emerging, such as source availability, time complexity, user context, and query drift, when operating with low-resource languages and multimedia data. From our analysis, we singled out a number of open issues and suggested some future directions of work: AI-driven and multilingual QE frameworks, real-time semantic search, and applications to diverse datatypes. These insights confirm the potential of semantic QE to respond to demands of growing complexity in information retrieval—meeting the objectives set forth in the introduction: theoretical insights with practical reflections toward further advances.

Additional Information and Declarations

Competing Interests

The authors declare that they have no competing interests.

Author Contributions

Azzah Allahim conceived and designed the experiments, performed the experiments, analyzed the data, performed the computation work, prepared figures and/or tables, and approved the final draft.

Asma Cherif conceived and designed the experiments, performed the experiments, analyzed the data, performed the computation work, authored or reviewed drafts of the article, and approved the final draft.

Abdessamad Imine conceived and designed the experiments, performed the experiments, analyzed the data, performed the computation work, authored or reviewed drafts of the article, and approved the final draft.

Data Availability

The following information was supplied regarding data availability:

This is a literature review.

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
