# Peer review of "Semantic approaches for query expansion: taxonomy, challenges, and future research directions"

_PeerJ Computer Science, doi:10.7717/peerj-cs.2664_

## Round 0.1 · original submission · Major Revisions

Dear authors,

Thank you for submitting your article. Reviewers have now commented on your article and suggest major revisions. We do encourage you to address the concerns and criticisms of the reviewers and resubmit your article once you have updated it accordingly. Especially necessary additions and modifications based on Reviewer 4 is important. When submitting the revised version of your article, it will be better to address the following:

1. A clearly defined research question should be addressed, comprehensive and systematic review of the literature should be conducted, and clearly reported, reproducible and systematic methods should be used to identify, select and critically appraise relevant research.
2. The Abstract misses to highlight “the need to write the review” although there are some survey and reviews.
3. The Introduction section should contain a well-developed and supported argument that meets the goals set out.
4. How this review paper will contribute to the scientific body of knowledge should be mentioned.
5. The coverage (both temporal and domain) of the literature and how the literature was distributed across time domains should be clearly provided.
6. The used specific filtering criteria specially to finalize the set of literature for review should be presented.
7. Many of the equations are part of the related sentences. Attention is needed for correct sentence formation.
8. Equations should be used with correct equation number. Please do not use “as follows”, “given as”, etc. Explanation of the equations should also be checked. All variables should be written in italic as in the equations. Their definitions and boundaries should be defined. Necessary references should be provided.

Best wishes,

Reviewer 1 ·

Basic reporting

This paper on Semantic Approaches for Query Expansion presents a comprehensive review of various techniques, methodologies, and recent advancements in the field. The well-structured paper begins with a thorough survey methodology, followed by discussions on related works, background information, query expansion categories, and different approaches (linguistic-based, ontology-based, and hybrid-based). The authors conducted an extensive literature review, covering a broad range of frameworks and techniques related to semantic query expansion. They highlighted the significance of semantic query expansion in addressing information retrieval system challenges. The paper's value is enhanced by the inclusion of recent high-quality works and the clear presentation of techniques in tabular form, providing a concise overview of the current state of the field.

Experimental design

No comment

Validity of the findings

No comment

Additional comments

The following comments need to be addressed:
• Lines 59-66: Apart from the reasons mentioned here, it is also important to mention that the existing literature surveys were done in or before 2019, which is like five years before. This itself should be a strong motivation for a new literature review paper.
• To enhance the discussion, include more comparative analysis between different semantic query expansion approaches, highlighting their effectiveness, applicability, and potential drawbacks. This can help readers understand the nuances of each technique better.:
• Enhance the discussion section by delving deeper into the implications of the findings presented in the paper. Offer insights into how the results contribute to the existing body of knowledge in semantic query expansion and information retrieval.

Reviewer 2 ·

Basic reporting

Strengths:
-Comprehensive overview: The paper provides a good overview of Semantic Query Expansion (SQE) by discussing different approaches, existing research, and future directions.
-Structured analysis: The breakdown of SQE approaches (linguistic structure, ontology, and hybrid) makes the information clear and easy to follow.
-Future-oriented: Highlighting areas like machine learning and multi-language support demonstrates the authors' understanding of the field's evolution.


Weaknesses:
-Limited details: While the paper covers various approaches, it might benefit from a deeper dive into the specific strengths and weaknesses of each SQE method with supporting examples.
-Lack of evaluation specifics: The review of existing research could be strengthened by including specific examples of successful or problematic implementations of SQE.
-Focus on challenges: While acknowledging the need for further research is valuable, the paper could also suggest potential solutions or ongoing efforts in overcoming those challenges.


General Comments:
-The writing seems grammatically correct and easy to understand.
-Consider adding a section on the impact or potential applications of SQE to make the research more relevant to a broader audience.

Recommendations for Improvement:
-Expand on specific approaches: Provide more details on how each SQE approach works, including its strengths and weaknesses with real-world examples.
-Strengthen evaluation section: Include specific examples of existing research on SQE, highlighting successful implementations and areas for improvement.

By incorporating these suggestions, the authors can strengthen the paper's overall impact and provide a more comprehensive understanding of Semantic Query Expansion.

Experimental design

no comment

Validity of the findings

The paper is primarily focused on reviewing and analyzing existing research on SQE approaches.

Reviewer 3 ·

Basic reporting

1-The manuscript is written in clear and professional English. Technical terms are appropriately used and explained.
Suggestions: There are a few minor grammatical issues that need attention to ensure clarity for an international audience. For instance, sentences can be streamlined for better readability. Example: "Its main purpose is to connect information based on its meaning." could be "It aims to connect information semantically."

2-The introduction provides a comprehensive overview of the topic, explaining the significance of semantic query expansion and its current research status.
Suggestions: The introduction could benefit from a more detailed explanation of the knowledge gap being filled by this review. This can help in setting a clearer context for the reader.

3-The literature cited is relevant and covers a broad range of studies within the field of query expansion.
Suggestions: Ensure that all references are up-to-date and cover the most recent advancements in the field. Some seminal works from the last couple of years may need to be included.

4-The manuscript follows a logical structure, with well-defined sections that are easy to follow.
Suggestions: Minor restructuring could improve clarity. For example, the "Contributions of the article" section could be integrated into the introduction for better flow.

5-The review is relevant and of broad interest to researchers in the fields of computer science and information retrieval.
Suggestions: Emphasize the interdisciplinary applications of semantic query expansion to attract a wider readership.

6-The review covers recent developments and provides a unique perspective on semantic query expansion.
Suggestions: Highlight the novel aspects and the unique contribution of this review compared to previous ones.

7-The subject and motivation are well introduced.
Suggestions: Enhance the clarity of the motivation by explicitly stating the gaps in existing literature that this review aims to address.

Experimental design

1-The content aligns well with the journal’s focus on computer science and information retrieval.
Suggestions: None.

2-The review demonstrates a thorough investigation of the topic.
Suggestions: Ensure that all methodologies are described in detail to maintain high technical standards.

3-The survey methodology is detailed and replicable.
Suggestions: Provide more details on the selection criteria for the reviewed frameworks to enhance replicability.

4-The survey methodology is comprehensive and covers a wide range of sources.
Suggestions: Discuss potential biases and how they were mitigated in the methodology section.

5-Sources are adequately cited throughout the manuscript.
Suggestions: Ensure that all direct quotes are properly formatted and referenced.

6-The review is logically organized into coherent sections.
Suggestions: Minor adjustments in the organization could improve readability, such as combining related subsections for better flow.

Validity of the findings

1-The manuscript does not focus on assessing the impact and novelty, which is appropriate for a literature review.
Suggestions: None.

2-The rationale and benefits to the literature are clearly stated, encouraging replication.
Suggestions: Provide more examples of successful applications of semantic query expansion to highlight its practical benefits.

3-The conclusions are well stated and linked to the research questions.
Suggestions: Ensure that the conclusions are strictly limited to the supporting results presented in the review.

4-The arguments are well-developed and supported by the reviewed literature.
Suggestions: Strengthen the link between the introduction and the conclusions to ensure the goals set out are fully met.

5-The conclusion identifies several unresolved questions and future research directions.
Suggestions: Provide more specific examples of potential future research directions to guide researchers.

Additional comments

The manuscript is well-written and provides a thorough overview of semantic query expansion techniques. The figures and tables are informative and well-integrated into the text.

Suggestions: Consider incorporating more visual aids, such as diagrams or flowcharts, to explain complex concepts. This will enhance understanding and engagement for the reader.

Reviewer 4 ·

Basic reporting

The manuscript overviews various query expansion methods, emphasising semantic approaches and reviewing frameworks developed between 2015 and 2024. It discusses the limitations of these methods and outlines the challenges in the field.
The authors distinguish the scope and contributions of the manuscript from other similar reviews. However, differences are not justified accordingly. The criteria for comparing against similar reviews, such as pipeline, approaches and challenges, are unjustifiable and must be defined accordingly. Furthermore, it is not fair to justify the uniqueness of this review in terms of up-to-date LR since the other reviews were conducted quite a few years ago. The authors must present a logical rationale for the distinction between this review and earlier reviews, particularly regarding the extent and demand of current advancements in IR techniques and artificial intelligence. In Table 1, the manuscript missed the crucial reference of:
Azad, H. K., & Deepak, A. (2019). Query expansion techniques for information retrieval: a survey. Information Processing & Management, 56(5), 1698-1735.
Carpineto, C., & Romano, G. (2012). A survey of automatic query expansion in information retrieval. ACM Computing Surveys (CSUR), 44(1), 1-50.

The organisation of the paper is peculiar, as the primary content merely provides a sequential overview of the investigations, while each of the categories are offered in the discussion part.
The background section provides too basic information about information retrieval, and some information is unnecessary for a state-of-the-art review paper. For example, the discussion on pre-processing, similarity measures, and evaluations is too rudimentary to be presented in this manuscript.

The categorisation of QE approaches is a bit confusing. The authors need to elaborate on the definitions and criteria when establishing the QE categories. The authors need to define what is their definition or criteria of semantic QE. The classification presented in the paper includes local-source, global-source and knowledge-based. However, the differences between the global-source and knwoledge-based are not very distinctive. Abdul-Nasir et al. (2019), for instance, categorised query expansion into local-context, global-context and knowledge-based, and asserted that the knowled-based method can be considered as part of the global method. In other words, some methods that use global sources can also be considered semantic QE.

Experimental design

The authors did not provide enough details about the study design. It is unclear which sources were used to collect the papers—whether it was the Web of Science, ACM Digital Library, or another database. Additionally, the exact inclusion criteria are not fully explained; they are mentioned in the form of keyword queries but lack further elaboration. The process of including papers should ideally involve at least two raters, with rater agreement calculated using a method like Cohen's kappa. It would also be helpful if the authors provided a spreadsheet listing the collected publications, indicating how each paper was selected and the reasons for its inclusion or exclusion.

Validity of the findings

The discussion of the findings lacks the necessary depth and critical analysis. Rather than offering insightful evaluations, it primarily presents a broad overview of the methods previously described. To strengthen the discussion, the authors should engage in a more distinguished critique, highlighting the strengths and weaknesses of each technique and exploring the implications of their findings.

The challenges highlighted in the discussion are indeed valuable; however, they appear disconnected from the scope and specific aspects of the Query Expansion (QE) techniques previously discussed. For instance, topics such as security considerations, threshold selection, and time complexity were not addressed in the earlier sections of the paper. This leaves readers questioning how the authors identified and arrived at these particular challenges. To enhance clarity and coherence, it would be beneficial for the authors to explicitly link these challenges to the methods and concepts covered in the discussion, ensuring that the progression of ideas is logical and well-supported.

Similar concerns for the future research directions of which the direction proposed was not tailored nicely to what has been discussed earlier.

Additional comments

QE has a long history in the IR field. The manuscript provides an essential review of semantic expansion. However, the review fails to establish the state-of-the-art of the area. Improvement of the manuscript is necessary to make the contributions of this manuscript valuable:
- The authors need to relook back at the classification of the semantic QE categories.
- The definition of each category needs to be elaborated and supported by formal and technical explanations.
- The discussion and future works must relate to the scope and directions of the discussed techniques.

---

## Round 0.2 · accepted · Accept

Dear Authors,

I would like to express my gratitude for the revised paper. It is important to note that one of the previous reviewers declined to review the revision, while two of them did not respond to the invitation in time. However, one reviewer who requested a major revision has accepted the paper. I have also conducted my own assessment of the revision and can confirm that I am satisfied with the current version. It is evident that the paper has undergone significant improvement and is now ready for publication.

Warm regards,

Reviewer 1 ·

Basic reporting

The review comments raised earlier have been addressed satisfactorily. The manuscript may be accepted for publication.

Experimental design

No comment

Validity of the findings

No comment